# Nanchangmycin regulates FYN, PTK2, and MAPK1/3 to control the fibrotic activity of human hepatic stellate cells

Wenyang Li[1,2†], Jennifer Y Chen[1,2‡§], Cheng Sun[1,2], Robert P Sparks[1,2], Lorena Pantano[3], Raza-Ur Rahman[1,2], Sean P Moran[1,2], Joshua V Pondick[1,2], Rory Kirchner[3], David Wrobel[2], Michael Bieler[4], Achim Sauer[4], Shannan J Ho Sui[3], Julia F Doerner[4], Jörg F Rippmann[4], Alan C Mullen[1,2,5]*

[1]Division of Gastroenterology, Massachusetts General Hospital, Boston, United States; [2]Harvard Medical School, Boston, United States; [3]Harvard T.H. Chan School of Public Health, Boston, United States; [4]Boehringer Ingelheim Pharma GmbH & Co, Biberach, Germany; [5]Harvard Stem Cell Institute, Cambridge, United States

**\*For correspondence:**
acmullen@mgh.harvard.edu

**Present address:** [†]College of Chemistry and Molecular Engineering, Peking University, Beijing, China; [‡]Department of Medicine, University of California, San Francisco, San Francisco, United States; [§]Liver Center, Department of Medicine, University of California, San Francisco, San Francisco, United States

**Abstract** Chronic liver injury causes fibrosis, characterized by the formation of scar tissue resulting from excessive accumulation of extracellular matrix (ECM) proteins. Hepatic stellate cell (HSC) myofibroblasts are the primary cell type responsible for liver fibrosis, yet there are currently no therapies directed at inhibiting the activity of HSC myofibroblasts. To search for potential antifibrotic compounds, we performed a high-throughput compound screen in primary human HSC myofibroblasts and identified 19 small molecules that induce HSC inactivation, including the polyether ionophore nanchangmycin (NCMC). NCMC induces lipid re-accumulation while reducing collagen expression, deposition of collagen in the extracellular matrix, cell proliferation, and migration. We find that NCMC increases cytosolic $Ca^{2+}$ and reduces the phosphorylated protein levels of FYN, PTK2 (FAK), MAPK1/3 (ERK2/1), HSPB1 (HSP27), and STAT5B. Further, depletion of each of these kinases suppress *COL1A1* expression. These studies reveal a signaling network triggered by NCMC to inactivate HSC myofibroblasts and reduce expression of proteins that compose the fibrotic scar. Identification of the antifibrotic effects of NCMC and the elucidation of pathways by which NCMC inhibits fibrosis provide new tools and therapeutic targets that could potentially be utilized to combat the development and progression of liver fibrosis.

## Editor's evaluation

The manuscript by Li et al., identifies the polyether ionophore nanchangmycin as a novel anti-fibrotic compound through a comprehensive chemical library screen. Given the lack of clinically available treatments for liver fibrosis, the anti-activation properties of nanchangmycin could represent a novel therapeutic avenue for the treatment of this disease. These studies pave way for future studies evaluating clinical efficacy of nanchangmycin and modified nanchangmycin compounds in the future.

## Introduction

Chronic liver disease and cirrhosis are the 11th leading cause of mortality in the United States, accounting for over 40,000 deaths annually (***Murphy et al., 2021***). Liver injuries, including those caused by viral infection, excessive alcohol intake, and nonalcoholic steatohepatitis, can lead to fibrosis, the accumulation of abnormal scar tissue, in the liver (***Bataller and Brenner, 2005***). If left unchecked, liver fibrosis can progress to cirrhosis and end-stage liver disease (***Bataller and Brenner,***

*2005*). HSC myofibroblasts are the primary cell type responsible for liver fibrosis (*Friedman et al., 1985*; *Mederacke et al., 2013*). HSCs reside in the perisinusoidal space and represent 5–8% of total cells in the liver. In their quiescent, nonfibrotic state, they store vitamin A as retinol ester in lipid droplets (*Geerts, 2001*). In response to chronic liver injury, HSCs are activated and trans-differentiate into HSC myofibroblasts, characterized by the loss of lipid droplets, increased contractility, and secretion of ECM proteins, leading to fibrosis (*Bataller and Brenner, 2005*; *Friedman, 2008*).

Resolution of liver fibrosis has been observed when the source of liver injury is removed, such as in patients with successful antiviral therapy against hepatitis B or C (*Benyon and Iredale, 2000*; *Bonis et al., 2001*; *Falize et al., 2006*). Two mechanisms can contribute to the reduction of activated HSC myofibroblasts during resolution of liver fibrosis – apoptosis of activated HSC myofibroblasts and reversion of HSC myofibroblasts to a more quiescent phenotype (*Friedman, 2008*). With regression of fibrosis, 40–50% of HSC myofibroblasts revert to an inactive state in vivo, which is associated with reduced collagen expression (*Kisseleva et al., 2012*; *Troeger et al., 2012*). These encouraging observations suggest that liver fibrosis is reversible and targeting HSC myofibroblasts to induce an inactive phenotype may serve as a therapeutic approach to treat patients with liver fibrosis.

Despite many efforts to understand HSC plasticity and target HSC myofibroblasts, there are currently no FDA-approved therapies directed at inhibiting the activity of HSC myofibroblasts. In our previous studies, we developed a small molecule screen to identify compounds that promote HSC inactivation (*Chen et al., 2017*). In a pilot screen, this approach revealed the antifibrotic effects of tricyclic antidepressants (TCAs). In mechanistic studies, we identified that TCAs inhibit the enzyme acid ceramidase (aCDase). In subsequent studies, we demonstrated that inhibiting aCDase regulates YAP/TAZ-mediated HSC inactivation and reduces fibrogenesis in mouse models and in human precision cut liver slices (*Alsamman et al., 2020*).

Here, we expanded our screen approximately 10-fold to include 15,867 experimental wells and developed a secondary screen to evaluate primary hits. We find that nanchangmycin (NCMC), a polyether ionophore, promotes HSC inactivation. Furthermore, we demonstrate that NCMC decreases proliferation, migration, and assembly of collagen fibers in the extracellular matrix. In additional mechanistic studies, we show that multiple kinases and signaling pathways are involved in mediating the impact of NCMC on HSC activities, including the FYN, PTK2 (FAK) and MAPK1/3 (ERK2/1) pathways. Taken together, this study defines NCMC as a potent antifibrotic compound that inactivates HSC myofibroblasts and highlights the FYN, PTK2, and MAPK1/3 pathways as potential downstream targets to inhibit liver fibrosis.

## Results

### A high-throughput small molecule screen identifies compounds that inactivate human hepatic stellate cell myofibroblasts

To identify small molecules that induce reversion of HSC myofibroblasts to an inactive phenotype, we screened 24 compound libraries consisting of 15,867 experimental wells using a high-throughput method to quantify lipid droplet accumulation as an indicator of HSC inactivation (*Chen et al., 2017*; *Figure 1A*, *Supplementary files 1and and 2*). Activated HSCs were seeded in 384-well plates, treated with compounds for 48 hr, fixed, and stained with Bodipy, a fluorescent lipid dye, to analyze the accumulation of lipid droplets. This approach allowed us to screen based on a feature characteristic of quiescent-like inactivated HSCs instead of limiting the readout to expression of a specific gene for the primary screen. Expression of *COL1A1* and *ACTA2* were then added as a secondary screen to focus on compounds that induced lipid accumulation and reduction of genes that mark the activation and fibrotic activity of HSCs. DMSO and nortriptyline were included as negative and positive controls, respectively on each plate. A scaled value was calculated for each experimental well based on the average percentage of Bodipy-positive cells, toxicity, and reproducibility and was normalized according to negative and positive controls on the same plate to minimize plate-specific effects.

Experimental wells with a scaled value higher than 0.85, a cutoff set as the top 10th percentile of nortriptyline-treated wells on the same plate, were defined as hits. To avoid losing potential hits due to plate-specific effects, the top three experimental wells with the highest scaled values on each plate were also included as hits even if their scaled values did not meet the 0.85 cutoff. A total of 711

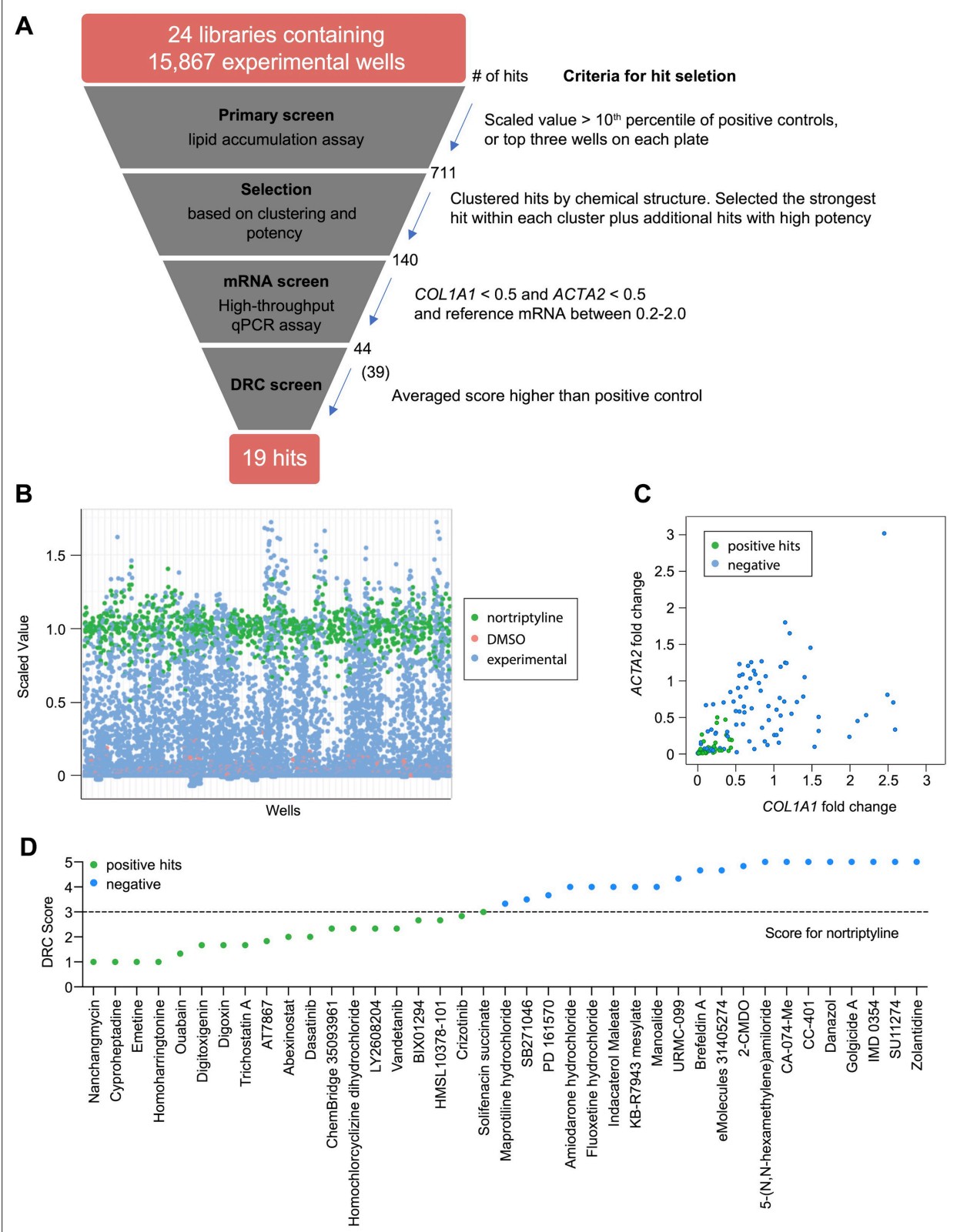

**Figure 1.** High-throughput small molecule screen in primary human HSCs.

(**A**) Overview of the small molecule screen. The number of candidate compounds (# of hits) is indicated at each step. The number in parenthesis represents the number of compounds tested in the final dose response curve (DRC) analysis due to availability. (**B**) Results of the primary lipid accumulation screen. Each dot indicates the mean scaled value of two replicates for each condition at 48 hr. Red dots represent negative control wells

*Figure 1 continued on next page*

*Figure 1 continued*

(DMSO), green dots represent positive control wells (nortriptyline, 27 µM), and blue dots represent experimental wells. (**C**) Results of the secondary mRNA screen. Each dot indicates the mean fold change of *ACTA2* and *COL1A1* after treatment with compounds (normalized to DMSO controls). *PSMB2* was used as the reference gene (n=4). Green dots represent positive hits (<0.5), and blue dots represent non-hits (negative). (**D**) Dose-response curves were plotted for 39 compounds and were scored by three researchers independently. The mean score for each compound was plotted. The dotted line indicates the score of the positive control nortriptyline. Green dots represent positive hits, and blue dots represent non-hits (negative). This figure has two supplements.

The online version of this article includes the following figure supplement(s) for figure 1:

**Figure supplement 1.** Selection of *PSMB2* as the reference mRNA for the qPCR-based secondary screening assay.

**Figure supplement 2.** Nineteen compounds were identified to induce lipid accumulation in HSCs.

experimental wells, containing 464 different compounds, met these criteria and were identified as primary hits (*Figure 1B* and *Supplementary file 3*).

To further narrow the candidate list for secondary screening, the 711 hits were separated into 102 clusters based on their chemical structure, with an average cluster size of seven compounds. The highest scoring hit with the most common structure within each cluster was selected as the representative compound for the cluster. Among the 102 representative compounds, ten compounds were removed because they contained pan assay interference structures (*Baell and Nissink, 2018*) or were themselves frequently identified as hits in screens. One compound was removed because it was a TCA, and we have previously demonstrated that TCAs target the sphingolipid pathway to inhibit HSC activity (*Chen et al., 2017*). One additional compound was removed because it had the same molecular formula as another selected hit (*Supplementary file 4*). In addition to representative compounds selected from each cluster, 50 compounds with high scaled values or promising structures were also selected. In total, 140 unique compounds were included in the secondary screen (*Figure 1A* and *Supplementary file 5*).

## Development of a secondary screening assay in primary human HSCs

Activated HSCs are characterized by increased expression of α-SMA (encoded by *ACTA2*) and type 1 collagen (encoded by *COL1A1*) (*Bataller and Brenner, 2005*; *Friedman, 2008*). We developed a high-throughput secondary assay to quantify *ACTA2* and *COL1A1* mRNA levels as indicators of HSC activity. HSCs were treated with compounds for 48 hr and then lysed for multiplexed qRT-PCR to quantify the house-keeping reference mRNA in the same well as *ACTA2* or *COL1A1*. Since a proper reference mRNA is critical for qRT-PCR based assays, we evaluated 18 housekeeping genes, consisting of seven commonly used genes as well as eleven reference genes identified from the literature (*Eisenberg and Levanon, 2013*). We first analyzed RNA sequencing data from HSCs under multiple conditions (*Chen et al., 2017*). Among these candidates, *GUSB*, *POLR2A*, *EMC7*, *VCP*, *PSMB2*, and *VPS29* showed the lowest standard deviation (0.15 or less). Further comparison of expression of these genes in inactivated HSCs (induced by the addition of nortriptyline or ceramide *Chen et al., 2017*) and culture-activated HSCs revealed that *GUSB*, *POLR2A*, *EMC7*, and *PSMB2* had the least fold change in expression (10% or less upon HSC inactivation). Thus, we selected these four reference mRNAs for further evaluation. *GAPDH*, which is used routinely as a reference control, was also included for comparison (*Figure 1—figure supplement 1A*). We quantified expression using qRT-PCR in HSC cDNA samples reverse-transcribed from equal amounts of total RNA. *PSMB2*, which encodes proteasome 20 S subunit beta 2, showed the least variation as indicated by standard deviation and was chosen as the reference mRNA for the secondary qRT-PCR-based screen (*Figure 1—figure supplement 1B*).

qRT-PCR was performed to quantify *ACTA2*, *COL1A1*, and *PSMB2* mRNA levels in each sample. Relative fold changes were calculated compared to DMSO control. We defined the following as criteria for compound advancement: 1. Fold change of *COL1A1* was reduced to less than 0.5 of DMSO control (FDR <0.05); 2. Fold change of *ACTA2* was reduced to less than 0.5 of DMSO (FDR <0.05); 3. Averaged *PSMB2* expression was between 0.2 and 2.0 of DMSO (*Figure 1C* and *Supplementary file 5*). This last criterion was added to avoid selecting compounds where large changes in *PSMB2* expression made it difficult to interpret changes in *ACTA2* and *COL1A1* expression. Of the 140 compounds, a total of 44 compounds met all three criteria. Five compounds were not commercially available, and 39 compounds were advanced for further analysis.

**Table 1.** Candidates from small molecule screening.

| Compound Name | DRC Score | Known function |
|---|:---:|:---:|
| Cyproheptadine * | 1.0 | Serotonin antagonist and antihistamine |
| Emetine * | 1.0 | Anti-protozoal, inhibitor of Zika and Ebola viruses |
| Homoharringtonine * | 1.0 | Translation elongation inhibitor |
| Nanchangmycin * | 1.0 | Polyether ionophore antibiotic, inhibitor of Zika virus |
| Ouabain * | 1.3 | Na/K-ATPase inhibitor |
| Digitoxigenin * | 1.7 | Na/K-ATPase inhibitor |
| Digoxin * | 1.7 | Na/K-ATPase inhibitor |
| Trichostatin A * | 1.7 | Histone deacetylase inhibitor |
| AT7867, HMSL10154-101-1 | 1.8 | Multi-kinase inhibitor |
| PCI-24781 (Abexinostat) * | 2.0 | Histone deacetylase inhibitor |
| Dasatinib | 2.0 | Multi-kinase inhibitor |
| ChemBridge 35093961 | 2.3 | IKK inhibitor |
| Homochlorcyclizine dihydrochloride | 2.3 | Antihistamine |
| LY2608204 | 2.3 | Glucokinase activator |
| Vandetanib | 2.3 | Multi-kinase inhibitor |
| BIX01294 (hydrochloride hydrate) | 2.7 | G9a histone methyltransferase inhibitor |
| HMSL10378-101 | 2.7 | Predicted to target GSK3B at 1 nM (ChEMBL) |
| Crizotinib | 2.8 | Multi-kinase inhibitor |
| Solifenacin succinate | 3.0 | Muscarinic receptor antagonist |

*Compounds with an EC50 less than 5 µM.

Next, we evaluated dose response curves (DRCs) for each compound at eight different concentrations, from 10 pM to 10 µM, using a Bodipy lipid accumulation assay similar to that employed in the primary screen. Dose response curves were scored blindly by three researchers (*Supplementary file 6* and Materials and methods), and nortriptyline served as a reference. Of the 39 compounds, 19 received an average score that was the same as or higher than nortriptyline controls (*Figure 1D*, *Figure 1—figure supplement 2*, and *Table 1*).

We then selected compounds for additional validation based on their EC50 and DRC scores to identify those considered the most potent. Of the nine compounds selected, two subgroups of compounds were identified based on similar bioactivity – histone deacetylase inhibitors (HDACIs), including trichostatin A and abexinostat, and Na/K-ATPase inhibitors, including ouabain, digitoxigenin, and digoxin. Histone deacetylases are linked to a variety of fibrotic disorders, including liver fibrosis (*Pang and Zhuang, 2010*). HDACIs, such as MC1568 and Valproate, have been reported to reduce HSC activation and alleviate liver fibrosis in animal models (*Yoon et al., 2019*). The presence of HDACIs in our final candidate list supports the validity of our screening approach in identifying potential liver fibrosis inhibitors. Na/K-ATPase activity may play a role in non-alcoholic fatty liver disease (*Sodhi et al., 2017*), but it is not clear how Na/K-ATPases regulate HSC activity and liver fibrosis. Due to the toxicity and narrow therapeutic dose range of cardiac glycosides, which limit their potential application in treatment of liver fibrosis, we decided not to pursue further evaluation of this group of compounds. Nanchangmycin (NCMC), a natural product of *Streptomyces nanchangensis*, is a polyether insecticidal antibiotic (*Sun et al., 2002*) and is one of the most potent hits. Studies of NCMC are limited, but it has been shown to have a broad spectrum of antiviral activity against diverse arboviruses (*Rausch et al., 2017*) and potentially SARS-CoV-2 infection (*Dittmar et al., 2020*; *Li et al., 2020*; *Svenningsen et al., 2020*). It also suppresses breast cancer stem cell activity and inhibits growth of breast cancer and multiple myeloma cells (*Huang et al., 2018*; *Xu et al., 2020*). The cellular targets

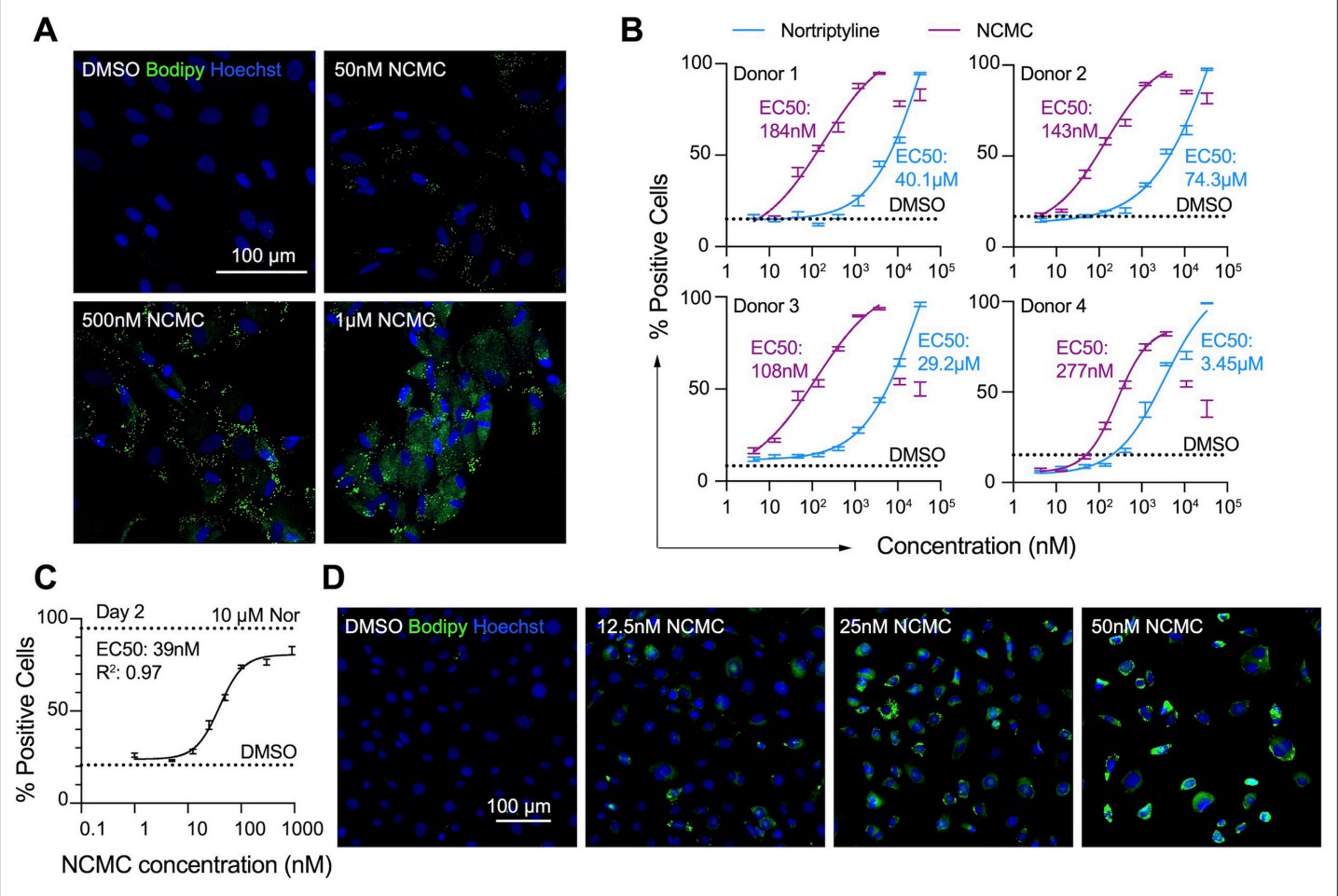

**Figure 2.** NCMC induces lipid accumulation in HSCs.

(**A**) Representative microscopic images of HSCs treated with DMSO and NCMC for 48 hr. Cells were stained with Bodipy to identify lipid droplets (green) and Hoechst to define nuclei (blue). Scale bar represents 100 μm. (**B**) Dose response curves for NCMC (purple) and nortriptyline (blue) in HSCs isolated from four different human donors at 48 hr. Dotted line represents the mean percentage of Bodipy-positive cells in DMSO control wells. Error bars represent mean ± SEM (n=12). One experiment was performed independently for each of four donor lines. Curves were generated by fitting the data to a sigmoidal model. The data from the highest two concentrations of NCMC treatment (11 and 33 μM) were not used for fitting due to higher toxicity at these concentrations, as indicated by cell number (**Figure 2—figure supplement 1**). (**C**) Dose response curve for NCMC treatment in murine primary HSCs at 48 hr. Dotted lines represent the averaged percentage of Bodipy-positive cells in DMSO-negative control wells (lower) and nortriptyline-positive control wells (10 μM, upper). Error bars represent mean ± SEM (n=6). Data are representative of three independent experiments. (**D**) Representative images of murine HSCs treated with DMSO and NCMC for 48 hr and stained with Bodipy to identify lipid droplets (green) and Hoechst to define nuclei (blue). Scale bar represents 100 μm. This figure has two supplements.

The online version of this article includes the following figure supplement(s) for figure 2:

**Figure supplement 1.** Effect of NCMC on cell number.

**Figure supplement 2.** Dose response curves of NCMC treatment at different time points in HSCs.

of NCMC remain largely unknown, and its effect on HSC activation and liver fibrosis has not been investigated. Therefore, we selected this compound to further validate its function in regulating HSC activity and explore its mechanism of action.

## Nanchangmycin induces lipid accumulation in both human and mouse primary HSCs

First, we confirmed the effect of NCMC on lipid accumulation in primary HSCs isolated from four human donors of different age, gender and race (donor information provided in Materials and methods). We observed that NCMC treatment significantly increased lipid droplet accumulation in

primary HSC lines compared to controls (*Figure 2A*), consistent with HSC inactivation. Compared to nortriptyline (*Chen et al., 2017*), NCMC exhibited higher potency in HSCs from all four donors, where 100–300 nM of NCMC exhibited similar effects on lipid droplet accumulation to 10 μM nortriptyline (*Figure 2B* and *Figure 2—figure supplement 1*). As we switched to NCMC from a different source with higher purity, we found that the new NCMC stock has a lower EC50 in HSCs from human donors and HSCs from C57BL/6 mice (*Figure 2C–D* and *Figure 2—figure supplement 2*). These results show that NCMC induces lipid accumulation in both human and murine HSCs.

## NCMC inhibits fibrotic gene expression in HSCs

We next quantified the effect of NCMC on *ACTA2* and *COL1A1* expression in multiple primary human HSC lines. NCMC treatment reduced both *ACTA2* and *COL1A1* levels at 100 nM or higher (*Figure 3A–B*). We also observed that NCMC reduced *Acta2* and *Col1a1* expression at multiple concentrations in murine HSCs at day 2 (*Figure 3C*). To investigate how NCMC affects the level of collagen deposited into the ECM, we performed the scar-in-a-jar assay to accelerate the process of ECM deposition with addition of molecular crowding reagents and TGF-β (*Chen et al., 2009*; *Good et al., 2019*). NCMC treatment significantly decreased collagen staining intensity and fiber area (*Figure 3D–E*). In addition to two-dimensional (2D) cell culture models, we also tested NCMC's effect on *COL1A1* expression in spheroids consisting of primary human HSCs and primary rat hepatocytes. Both the basal expression of *COL1A1* and TGF-β-induced *COL1A1* expression were significantly reduced by NCMC treatment (*Figure 3F*).

HSCs were next treated with NCMC and analyzed by RNA-sequencing analysis, which revealed that NCMC broadly affects genes associated with fibrosis. Among the top gene sets negatively enriched in the NCMC-treated group were ECM-related signatures, including ECM structural constituent and collagen-containing ECM, as well as signatures relevant to migration, including contractile fibers. Of note, genes associated with oligosaccharide lipid intermediate biosynthetic process were positively enriched, possibly contributing to the re-accumulation of lipid droplets (*Figure 3G* and *Supplementary file 7*). We compared the RNA sequencing data with a canonical HSC gene signature (*Zhang et al., 2016*), an HSC-specific signature that is highly and uniquely expressed in HSCs and correlates with the extent of fibrosis (*Zhang et al., 2016*), and the liver cirrhosis signature from Disgenet database (*Piñero et al., 2020*). We observed that these signatures were significantly negatively enriched (*Figure 3—figure supplement 1* and *Supplementary file 8*). Visualization of the canonical HSC gene signature (*Zhang et al., 2016*) also demonstrated that many genes that promote liver fibrosis, including those that encode collagens, TGF-β, and PDGF pathway components, are inhibited by NCMC (*Figure 3H* and *Supplementary file 8*). The mRNA level of the classic quiescent or inactivated HSC marker PPARG (PPARγ) is also induced with NCMC treatment (*Figure 3—figure supplement 2*). Taken together, these data suggest that NCMC triggers a global change in HSC gene expression, resulting in a decrease in profibrotic activity. We also used the Enrichr online pathway analysis tool (*Chen et al., 2013*; *Kuleshov et al., 2016*; *Xie et al., 2021*) to analyze the 1904 genes that are significantly (FDR <0.05) upregulated greater than two fold and the 2,620 genes that are significantly (FDR <0.05) downregulated more than twofold by NCMC (*Supplementary file 9*). The results from analysis of five databases were compared, including Reactome 2016, MSigDB Hallmark 2020, WikiPathway 2021 Human, Elsevier Pathway Collection, and KEGG 2021 Human (*Figure 3—figure supplements 3 and 4* and *Supplementary file 10*). Among the 30 pathways significantly enriched (FDR <0.05) for NCMC-upregulated genes, the unfolded protein response, sterol regulatory element-binding proteins (SREBP) signaling and cholesterol synthesis were identified through analysis of multiple databases. Among the 270 pathways significantly enriched (FDR <0.05) for NCMC-downregulated genes, a few pathways were identified to be represented more than once, for example, TGF-β signaling, $Ca^{2+}$ response and regulation, interferon response, estrogen response, and focal adhesion-PI3K/Akt signaling pathway. These pathways may contribute to the HSC-inactivating effects of NCMC.

## NCMC reduces migration and proliferation of HSCs

In addition to secretion of ECM proteins, activated HSCs demonstrate enhanced migration capabilities (*Hernandez-Gea and Friedman, 2011*). Thus, we performed transwell migration and scratch wound healing assays to evaluate how NCMC affects HSC migration. HSCs were pre-treated with NCMC for two days before seeding in cell culture inserts with permeable membranes. After 6–24 hr, HSCs that

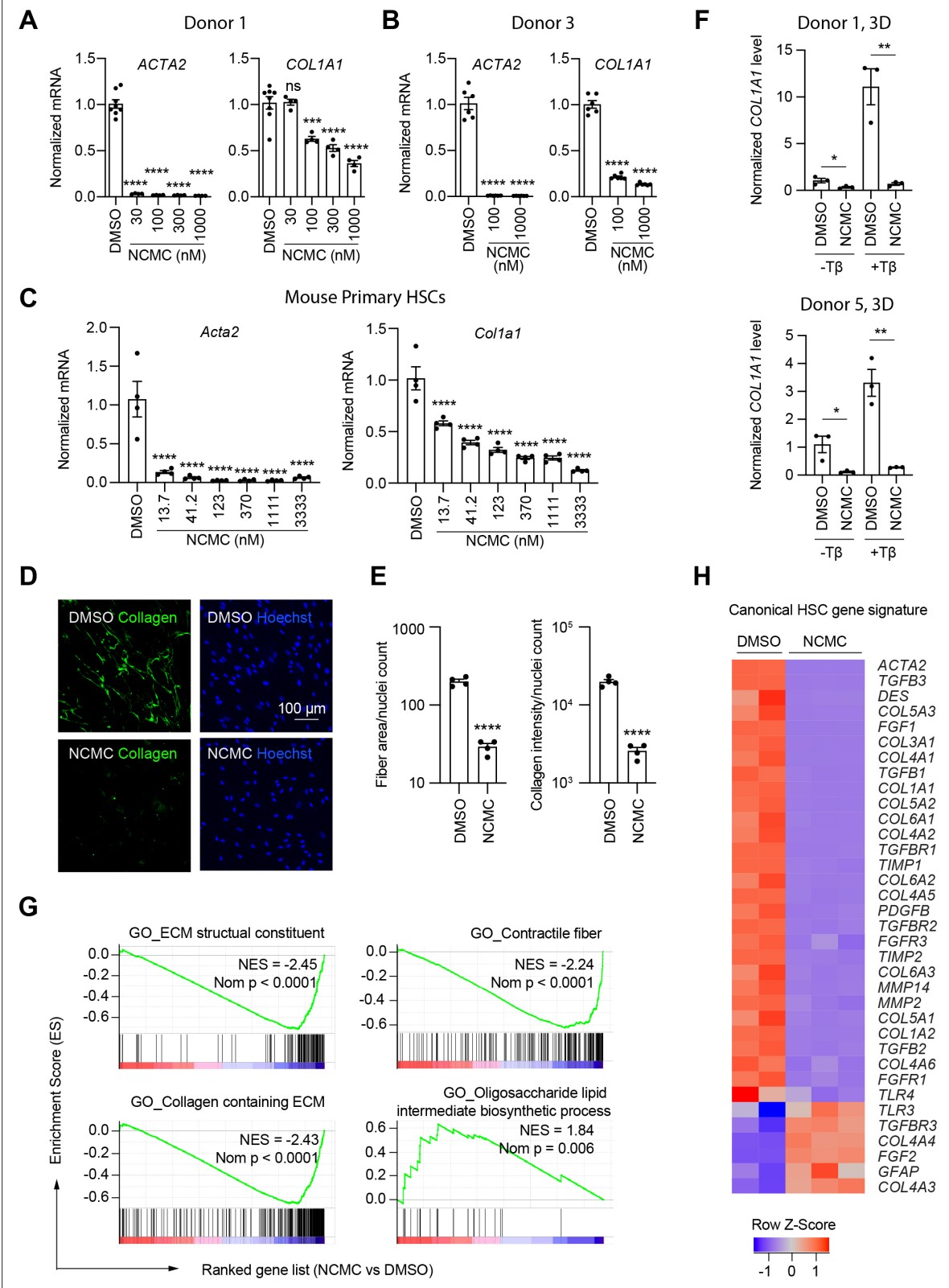

**Figure 3.** NCMC inhibits expression of fibrotic genes in HSCs. (**A-B**) Effect of 48 hr NCMC treatment on *ACTA2* and *COL1A1* in HSCs from human donors 1 (**A**) and 3 (**B**). Error bars represent mean ± SEM (n=3). Data are representative of three independent experiments. ns indicates not significant, *** indicates p<0.001, and **** indicates p<0.0001 (one-way ANOVA test). (**C**) Effect of 48 hr NCMC treatment on *Acta2* and *Col1a1* in primary mouse HSCs. Error bars represent mean ± SEM (n=4). Data are representative of three independent experiments. **** indicates p<0.0001 (one-way ANOVA

*Figure 3 continued on next page*

*Figure 3 continued*

test). (**D–E**) Effect of 48 hr NCMC treatment (1 μM) on collagen deposition in ECM. (**D**): representative images. Scale bar represents 100 μm. Collagen protein is indicated in green and nuclei for the same field are indicated in blue. (**E**): quantified results. Error bars represent mean ± SEM (n=4). Data are representative of three independent experiments. **** indicates p<0.0001 (Student's t-test). (**F**) qPCR analysis of *COL1A1* in HSC-hepatocyte spheroids treated with NCMC with and without TGF-β (Tβ). Error bars represent mean ± SEM (n=3). One experiment was performed independently for each donor shown. * indicates p<0.05 (Student's t-test) and ** indicates p<0.01 (Student's t-test). Analysis was performed on day 3 (3D). (**G–H**) RNA sequencing analysis of HSCs (donor 1) treated with DMSO or 1 μM NCMC for 48 hr. (**G**) Representative gene sets from the gene set enrichment analysis (GSEA). NES refers to normalized enrichment score. Nom P refers to Nominal P value. Vertical black lines refer to affected genes in the indicated signatures. Red color indicates positive correlation, and blue color indicates negative correlation. (**H**): Heatmap showing RNA-seq expression for the canonical HSC gene signature (***Zhang et al., 2016***). All genes from the signature that are expressed in HSCs (with a minimum of five reads) were shown regardless of their expression patterns. Z-score values are also provided in ***Supplementary file 8***. This figure has three supplements.

The online version of this article includes the following figure supplement(s) for figure 3:

**Figure supplement 1.** Gene set enrichment analysis results of HSC (***Zhang et al., 2016***) and cirrhosis (***Piñero et al., 2020***) gene signatures.

**Figure supplement 2.** NCMC treatment increases *PPARG* mRNA level.

**Figure supplement 3.** Pathway analyses of genes significantly upregulated in response to NCMC.

**Figure supplement 4.** Pathway analyses of genes significantly downregulated in response to NCMC.

migrated through the membrane were stained and counted. NCMC treatment significantly reduced the number of cells that migrated through the membrane compared to DMSO controls (***Figure 4A***).

We also examined the wound healing capability of HSCs. HSCs were seeded onto plates containing inserts that block cells from accessing and attaching to a strip at the bottom of the well. After removal of the insert, DMSO or 1 μM NCMC was added, and HSCs filled the gap or 'wound' field through migration and proliferation. After 30 hr of treatment, the DMSO-treated control cells closed the gap, whereas the gap remained for NCMC-treated cells (***Figure 4B***). While inhibition of HSC migration by NCMC is likely the major contributor to the difference observed between NCMC treatment and controls at shorter time points, such as 6 hr for the transwell assay, the difference at longer time points could be attributed to reductions in both migration and proliferation.

We next determined how NCMC affects HSC proliferation. We treated HSCs isolated from two human donors with DMSO control and increasing concentrations of NCMC and counted Hoechst-stained nuclei on five consecutive days. NCMC treatment reduced cell proliferation at concentrations of 25 nM and higher, and this effect on proliferation was first evident on day 2 (***Figure 4C***). A fraction of HSCs undergo apoptosis with the removal of fibrotic stimuli in vivo (***Kisseleva et al., 2012***), and we evaluated the contribution of apoptosis to the effect of NCMC on day 2. Based on the analysis of Annexin V and propidium iodide (PI) by flow cytometry, NCMC treatment only showed a small increase in the percentage of apoptotic cells in one of two donor HSC lines at 1 μM and no increase at 100 nM. There was also a small increase in necrotic cells in NCMC treatment compared to controls (***Figure 4D***). In addition, analysis of the proliferation marker Ki-67 and PI showed that NCMC treatment increased the percentage of G0 quiescent cells (***Figure 4E*** and ***Figure 4—figure supplement 1***). Taken together, these results suggest that HSCs enter a more quiescent-like, non-proliferative state upon NCMC treatment, and apoptosis was only observed in a small fraction of cells.

## Modulation of Ca²⁺ signaling may contribute to the effect of NCMC on HSC inactivation

NCMC is a polyether ionophore and has been shown to increase cytosolic $Ca^{2+}$ in a cancer cell line (***Huang et al., 2018***). We examined the activity of NCMC as a calcium ionophore in HSCs. We loaded HSCs with a fluorescent $Ca^{2+}$ indicator (Fluo-4 NW) and treated cells with NCMC. Ionomycin and thapsigargin were included as positive controls (***Jones and Sharpe, 1994***; ***Morgan and Jacob, 1994***). At 10 μM, NCMC increased cytosolic $Ca^{2+}$ as did ionomycin and thapsigargin (***Figure 4F***). We also performed the same analysis with ethylene glycol tetraacetic acid (EGTA) to chelate $Ca^{2+}$ and eliminate any contribution from extracellular $Ca^{2+}$ during the assay. Similar to the effect observed with thapsigargin (***Ribeiro et al., 2018***), the increase of cytosolic $Ca^{2+}$ in response to NCMC was not sensitive to EGTA (***Figure 4—figure supplement 2A***), suggesting that the immediate increase of cytosolic $Ca^{2+}$ following NCMC treatment is due to release of calcium from intracellular stores. Analysis of a

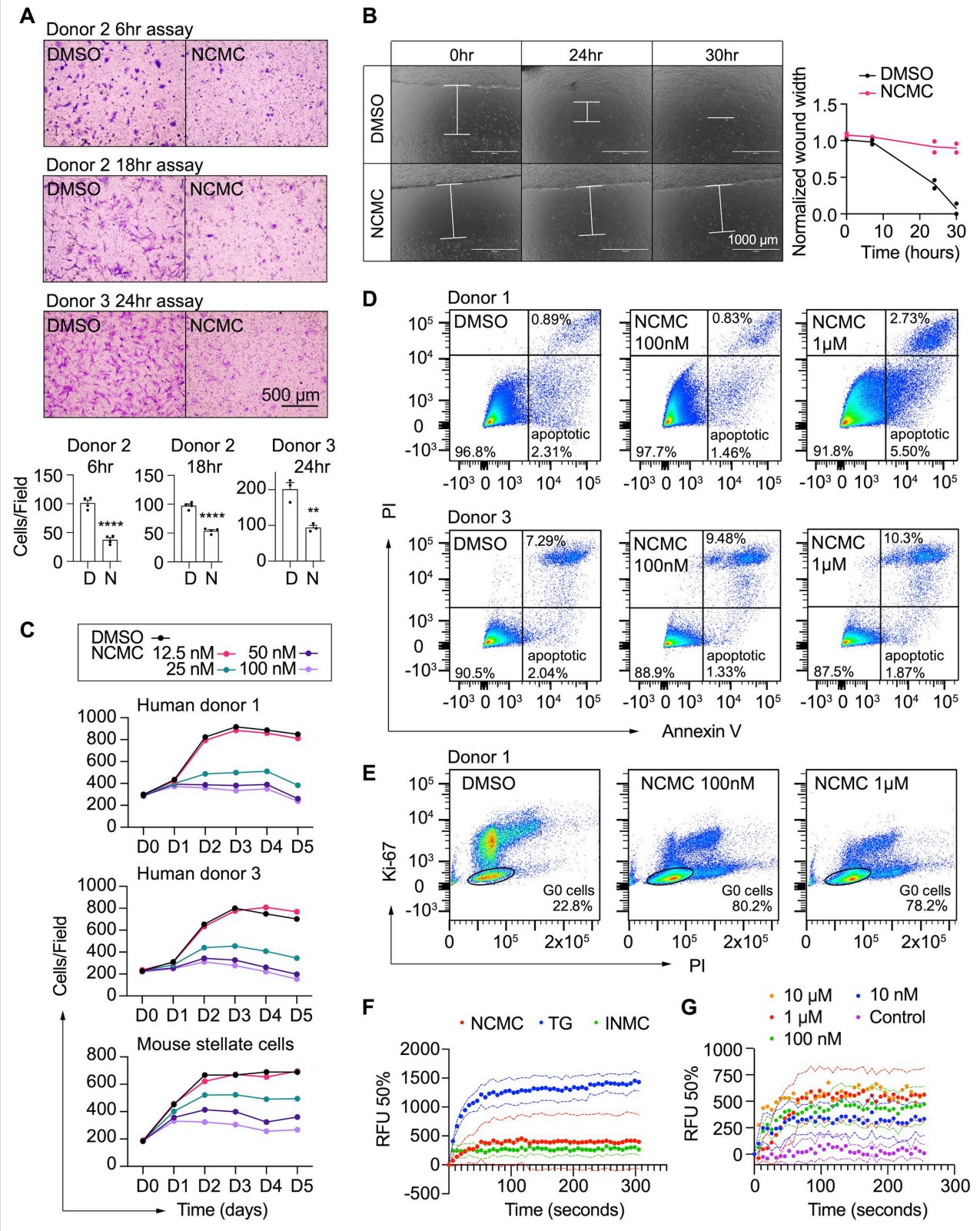

**Figure 4.** NCMC inhibits HSC migration and proliferation and increases cytosolic calcium concentration.
 (**A**) Transwell migration assay results of HSCs treated with DMSO or 1 µM NCMC for 48 hr. Top: representative images. Scale bar represents 500 µm. Bottom: quantification of migrated cells (n=3 for each experiment). ** indicates p<0.01, and **** indicates p<0.0001 (Student's t-test). (**B**) Wound healing assay results of HSCs treated with DMSO or NCMC. HSCs were seeded in complete medium, and immediately after generating the wound

*Figure 4 continued on next page*

*Figure 4 continued*
field, DMSO and 1 µM NCMC were added. The closure of the wound field was monitored for up to 30 hr as indicated. Left: representative images. White bars highlight the width of the wound field. Scale bar represents 1000 µm. Right: quantification of wound width. (n=2). Data are representative of three independent experiments. (**C**) Cell count for HSCs treated with DMSO or NCMC over the indicated time in days. Top: human HSCs from donor 1. Middle: human HSCs from donor 3. Bottom: mouse HSCs. Error bars represent mean ± SEM (n=6) but are too small to be visualized. One experiment was performed independently for each HSC line shown. (**D**) Flow cytometry analysis of Annexin V and propidium iodide (PI) stained HSCs from human donor 1 (top) and 3 (bottom) treated with DMSO or NCMC for 48 hr. Plots are representative of two independent experiments. (**E**) Flow cytometry analysis of Ki-67 and PI stained HSCs from human donor 1 treated with DMSO or NCMC for 24 hr. Plots are representative of two independent experiments. (**F**) Measurement of cytosolic calcium level using fluo-4 NW. HSCs from donor 3 were pre-loaded with fluo-4 NW and fluorescent intensity was read immediately after adding compounds (NCMC: nanchangmycin [red], TG: thapsigargin [blue], INMC: ionomycin [green]). The plot demonstrates results from three independent experiments. Solid dots represent mean, and dotted lines represent SEM (n=3). RFU: Relative fluorescence unit. (**G**) Measurement of cytosolic calcium level after adding NCMC at indicated concentrations in HSCs from donor 3 (magenta: control (no compound, no DMSO), blue: 10 nM, green: 100 nM, red 1 µM, orange: 10 µM). The plot demonstrates results from one experiment for 10 µM and five independent experiments for the other concentrations. Solid dots represent mean, and dotted lines represent SEM (n=3). This figure has two supplements.

The online version of this article includes the following figure supplement(s) for figure 4:

**Figure supplement 1.** Flow cytometry analysis of Ki-67 and PI stained HSCs.

**Figure supplement 2.** Measurement of cytosolic calcium level using fluo-4 NW.

dose response of NCMC demonstrated an increase in cytosolic $Ca^{2+}$ at concentrations as low as 10 nM (*Figure 4G* and *Figure 4—figure supplement 2B*).

## NCMC reduces *COL1A1* expression in HSCs through the FYN pathway

Calcium signaling regulates mitogen-activated protein kinases and non-receptor tyrosine kinases (*Filvaroff et al., 1990*; *Rusanescu et al., 1995*; *Xia et al., 1996*), and we analyzed a kinase array to define kinase signaling molecules modulated by NCMC. HSCs were treated with DMSO or 1 µM NCMC for 1 and 18 hours (*Figure 5A*). Among the 45 proteins tested, FYN phosphorylation at Y420 was reduced by approximately 40% at both 1 hr and 18 hr. We selected HSPB1 (HSP27), MAPK1/3 (ERK2/1), STAT5A/B, and PTK2 (FAK) to study further in addition to FYN because (1) they also showed decreased phosphorylation at 18 hr, and (2) genes encoding these products are expressed at a relatively high level in HSCs, as indicated from RNA sequencing data, suggesting that these may also be potential mechanistic targets of NCMC in HSCs.

To further investigate the role of these seven kinases in human HSCs, we depleted each kinase using pooled siRNAs in human HSCs from three donors. We observed a consistent reduction of *COL1A1* with depletion of *FYN, HSPB1, MAPK1, MAPK3,* and *STAT5B* (*Figure 5B–C*, *Figure 5—figure supplement 1*), suggesting that each kinase contributes to regulation of *COL1A1* expression.

Among the kinases that consistently reduced *COL1A1* expression, the reduction of FYN phosphorylation at Y420 was the most prominent following one hour of NCMC treatment, indicating that FYN may mediate the immediate response to NCMC in HSCs. Therefore, we further investigated FYN activity. Western blot of FYN revealed two bands – the upper band was reduced in HSCs treated with NCMC, while the lower band showed little change (*Figure 6—figure supplement 1A*). Both bands were reduced with depletion of FYN (*Figure 6—figure supplement 2*), suggesting that both products are encoded by *FYN* mRNA. FYN phosphorylation was not directly evaluated because antibodies that uniquely recognize phosphorylated FYN are not available. We also probed with a phospho-Src family antibody, which recognizes phosphorylated FYN and other Src family proteins (*Figure 6—figure supplement 1B*). Two bands of approximately the same size are observed with FYN antibody and pSrc antibody, suggesting that both bands may represent phosphorylated FYN, while it is the product in the upper band that is affected by NCMC treatment. qPCR analysis of NCMC-treated HSCs showed that *FYN* mRNA level was not affected (*Figure 6—figure supplement 3*), further indicating that NCMC regulates FYN through a post transcriptional mechanism.

Both depletion of FYN using two different siRNA duplexes (*Figure 6A–B*) and treatment with 1-Naphthyl PP1, an inhibitor of v-Src, FYN, and ABL (*Figure 6C*), significantly reduced *COL1A1* mRNA level in HSCs. Collagen deposition in the ECM was also impaired by FYN depletion, as indicated by the reduced collagen intensity and fiber area in the scar-in-a-jar assay (*Figure 6D*). In addition, ectopic expression of a dominant negative Y213A FYN mutant (*Kaspar and Jaiswal, 2011*) reduced *COL1A1*

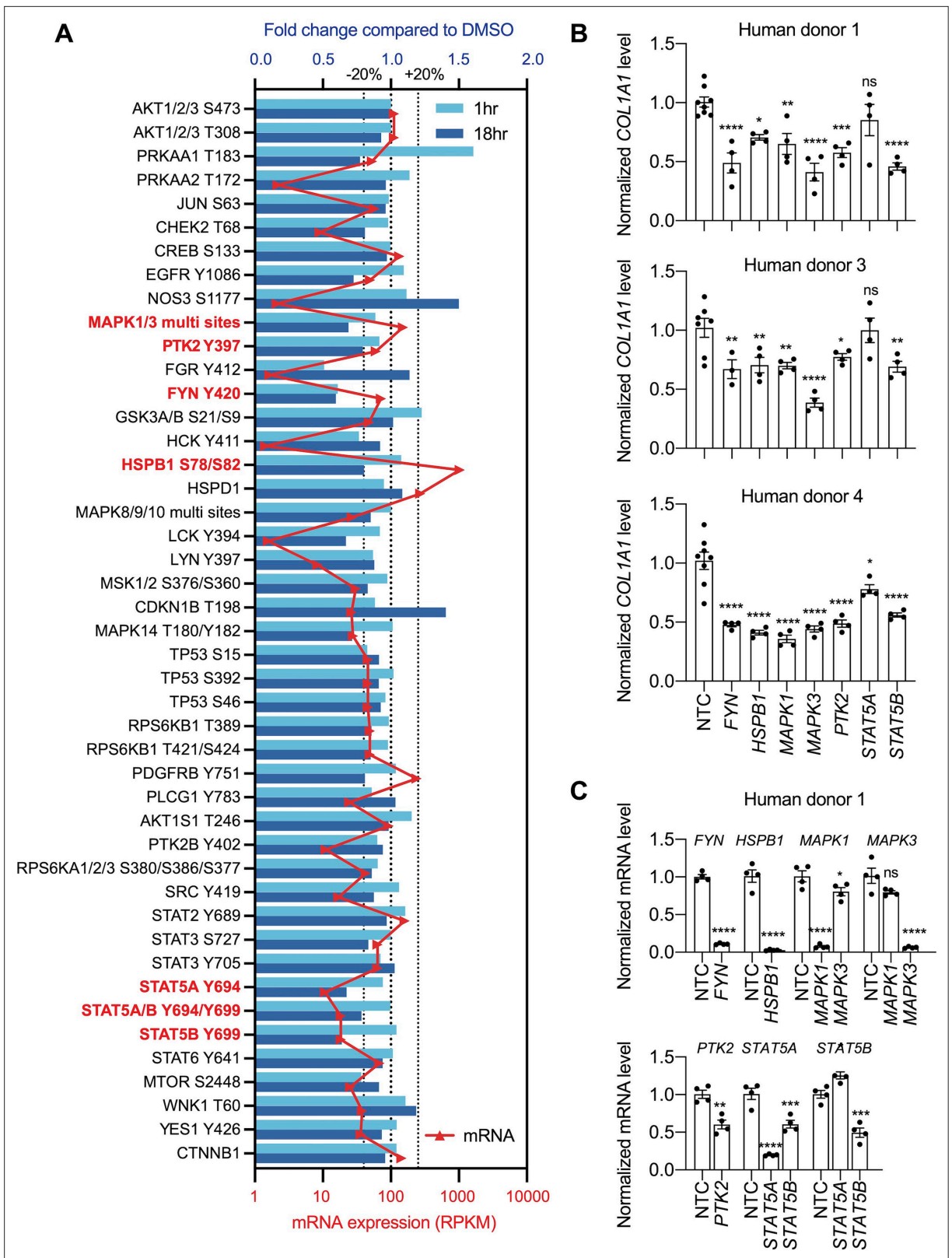

**Figure 5.** Multiple kinases mediate the effect of NCMC on *COL1A1* expression.

(**A**) Kinase array analysis of HSCs treated with DMSO or 1 µM NCMC for 1 or 18 hr. Blue bars indicate mean fold change (n=2) in phosphorylation at specified sites in NCMC-treated cells compared to DMSO-treated cells at 1 hr (light blue) or 18 hr (dark blue). Red triangles indicate the mean RPKM of each corresponding kinase mRNA based on RNA sequencing of HSCs (***Chen et al., 2017***). The three dotted lines represent 20% increase/decrease or

*Figure 5 continued on next page*

*Figure 5 continued*

no change in phosphorylation. Kinases highlighted in red were chosen for further investigation. (**B**) The expression of each candidate kinase gene was depleted using pooled siRNAs, and after 72 hr, *COL1A1* level was determined by qRT-PCR in HSCs isolated from human donor 1 (top), 3 (middle), or 4 (bottom). A non-targeting siRNA is used as a control (NTC). Error bars represent mean ± SEM (n≥4, as indicated by the number of dots). ns indicates not significant (p>0.05), * indicates p<0.05, ** indicates p<0.01, *** indicates p<0.001, and **** indicates p<0.0001 (one-way ANOVA test). (**C**) Knockdown efficiency of each siRNA pool in HSCs from human donor 1. Error bars represent mean ± SEM (n≥4). ns indicates not significant (p>0.05), * indicates p<0.05, ** indicates p<0.01, *** indicates p<0.001, and **** indicates p<0.0001 (Student's t-test performed for *FYN*, *HSPB1*, and *PTK2* depletion, and one-way ANOVA test performed for *MAPK1*, *MAPK3*, *STAT5A*, and *STAT5B* depletion). One experiment was performed independently for each HSC line shown. This figure has one supplement.

The online version of this article includes the following figure supplement(s) for figure 5:

**Figure supplement 1.** Knockdown efficiency of each siRNA pool in HSCs from human donor #3 (top) and donor #4 (bottom) compared to a non-targeting siRNA control (NTC).

expression in HSCs but did not further decrease *COL1A1* level in NCMC-treated cells, suggesting that FYN inhibition is likely to be downstream of NCMC (*Figure 6E*). Lastly, expression of dominant negative FYN in HSCs resulted in a decrease in phosphorylated MAPK (*Figure 6F*), suggesting that FYN may crosstalk with the MAPK pathway to exert its function.

## MAPK1/3 and PTK2 regulate HSC migration

Next, we asked if FYN regulates other HSC phenotypes that are observed with NCMC treatment. HSCs were transfected with pooled FYN siRNAs, and after 3 days, cells were seeded for transwell migration assay. As PTK2 is known to regulate migration of lung fibroblasts (*Zhao et al., 2016*) and HSCs (*Zhao et al., 2017*), we included *PTK2* siRNAs as a positive control. While PTK2 depletion consistently suppressed migration of HSCs isolated from three different human donors, FYN depletion only reduced migration in one HSC line (*Figure 7A–B*). We also observed that dominant negative FYN promotes phosphorylation of PTK2 (*Figure 6F*), while NCMC reduces phosphorylation of PTK2 (*Figure 6—figure supplement 4*), suggesting that NCMC controls PTK2 phosphorylation and HSC migration through a pathway that is independent of FYN.

We further tested how other kinases affected by NCMC (MAPK1/3 and HSPB1) regulate HSC migration. Depletion of MAPK1/3 consistently reduced migration across different donors, whereas the influence of HSPB1 depletion varied across HSCs from different donors (*Figure 7C*). These data indicate that NCMC regulates HSC migration through multiple downstream signaling pathways likely targeting PTK2 and MAPK1/3 as the primary paths to inhibit migration.

## Pharmacokinetic studies of NCMC

We next sought to determine whether NCMC itself could serve as an antifibrotic compound in vivo. Previous studies showed that dosing of 1 mg/kg every other day via intraperitoneal (IP) injection for three weeks resulted in decreased breast cancer tumor size (*Huang et al., 2018*), and that oral administration of 2 mg/kg or 4 mg/kg NCMC daily for 20 days reduces multiple myeloma subcutaneous xenografts in nude mice without affecting liver functions, as indicated by aspartate aminotransferase (AST), alkaline phosphatase (ALP), and alanine aminotransferase (ALT) levels (*Xu et al., 2020*). To estimate the dose necessary to achieve NCMC exposure in the liver at which an antifibrotic response might be observed, we initially performed pharmacokinetic (PK) studies following IP injection of NCMC at 1 mg/kg to mice. We found a relatively flat plasma concentration-time profile of NCMC after a single dose of 1 mg/kg. The maximum plasma concentration ($C_{max}$) of 108±39.7 nM was reached at 4 hr followed by a decrease to less than 50 nM after 12 hr. Based on the analysis of plasma concentration over time, we estimated a terminal elimination half-life ($t_{1/2}$) of 5.7±0.87 hr. As anticipated from the flat concentration-time curve, the calculated volume of distribution (Vss/F) is nearly 10 L/kg, indicating substantial distribution into tissue.

We next repeated IP dosing daily at 1 mg/kg for 4 days to determine how this affects plasma and tissue concentrations. The plasma concentration-time profile measured after the fourth dose was comparable to that after single dose, giving no hint of an accumulation of the compound after multiple dosing (*Table 2*). We detected 10-fold higher NCMC concentration in fat tissue compared to plasma, whereas it was undetectable in liver and muscle. The trough total plasma concentration was in the range of EC50 in cell culture, which would result in a lower effective concentration when taking

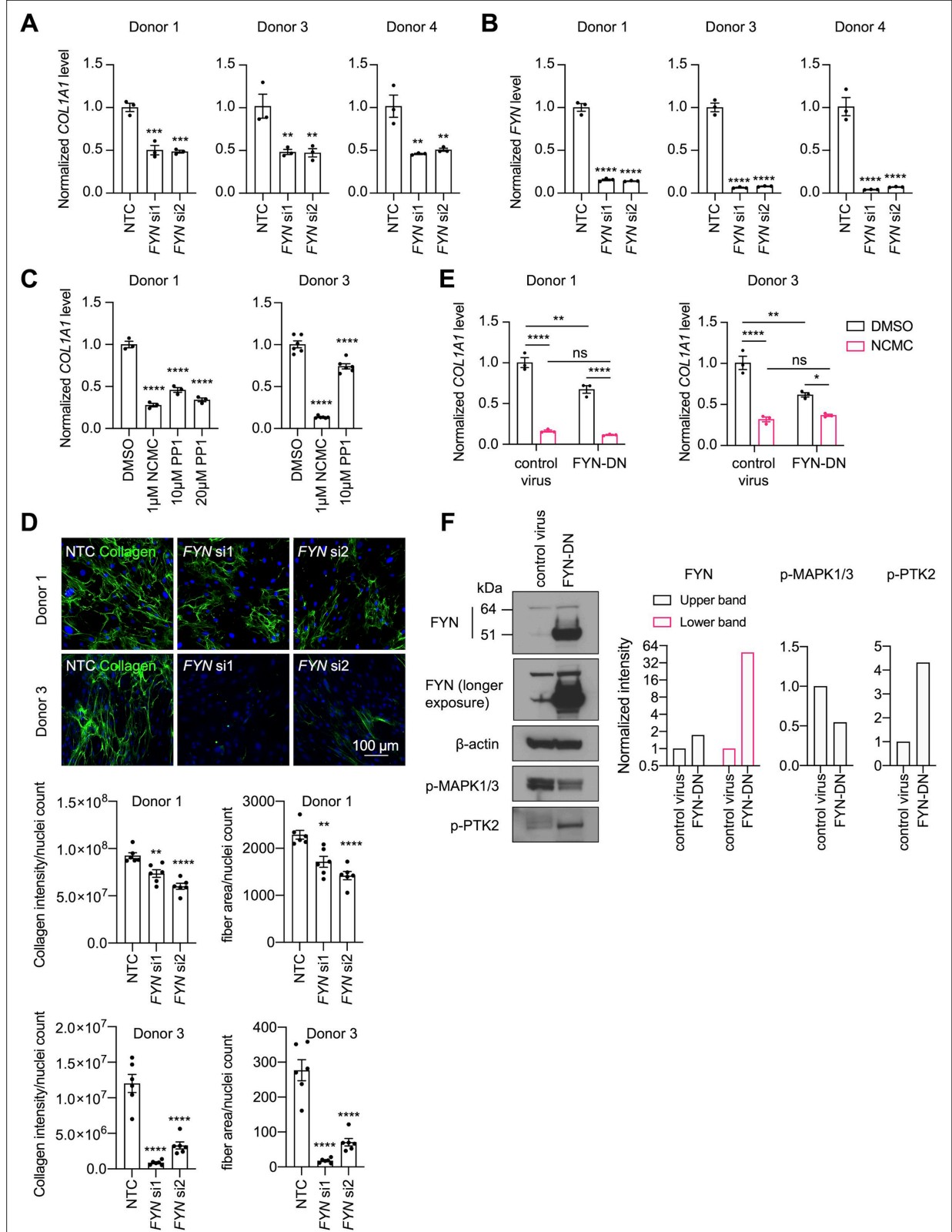

**Figure 6.** FYN/MAPK pathway regulates collagen expression.

(**A-B**) *FYN* was depleted in HSCs with two siRNAs (si1 and si2). The expression levels of *COL1A1* (**A**) and *FYN* (**B**) were analyzed by qPCR after 72 hr in comparison to a non-targeting control siRNA (NTC). Error bars represent mean ± SEM (n=3). At least one experiment was performed independently for each of three donor lines. ** indicates p<0.01, *** indicates p<0.001, and **** indicates p<0.0001 (one-way ANOVA test). (**C**) HSCs were treated with

*Figure 6 continued*

NCMC or 1-Naphthyl PP1 (PP1) for 48 hr. *COL1A1* level was analyzed by qPCR. Error bars represent mean ± SEM (n=3 for donor 1, and n=6 for donor 3). Data are representative of three independent experiments for donor 1 and experiment for donor 3. **** indicates p<0.0001 (one-way ANOVA test). (**D**) Effect of FYN-depletion on collagen deposition in ECM. Top: representative images. Scale bar represents 100 μm. Bottom: quantified results. Error bars represent mean ± SEM (n=6). Data are representative of two independent experiments. ** indicates p<0.01, and **** indicates p<0.0001 (one-way ANOVA test). (**E**) HSCs transduced with control virus or virus containing the cDNA encoding dominant negative mutant FYN (FYN-DN) were treated with DMSO or 100 nM NCMC for 48 hr. Expression of *COL1A1* was quantified by qPCR. Error bars represent mean ± SEM (n=3). Data are representative of three independent experiments. ns indicates not significant (p>0.05), * indicates p<0.05, ** indicates p<0.01, and **** indicates p<0.0001 (two-way ANOVA test). (**F**) Phospho-MAPK and phospho-PTK2 levels were determined by western blot in control HSCs and HSCs overexpressing DN-FYN. Left: representative Western blot results. Right: quantified results. Representative of two independent experiments. This figure has four supplements.

The online version of this article includes the following figure supplement(s) for figure 6:

**Figure supplement 1.** Western blot of HSCs treated with 1 μM NCMC for 18 hr.

**Figure supplement 2.** Western blot for FYN.

**Figure supplement 3.** *FYN* mRNA level was not affected by NCMC treatment.

**Figure supplement 4.** NCMC decreases PTK2 phosphorylation.

into consideration the fraction of NCMC bound to serum proteins, which we measured at >99% (the reported plasma concentrations refer to the total = free + bound plasma concentrations). Further, hepatic microsomes from mice showed evidence of degradation of NCMC by phase I liver metabolism, as indicated by the short half-life (37.8 min) and high derived intrinsic clearance (36.7 μl/min/mg protein).

In summary, the pharmacokinetic data suggest that NCMC will not reach sufficient exposure in the liver to achieve hepatic antifibrotic effects in vivo.

## Discussion

Liver fibrosis is the major driver of liver failure in all etiologies of chronic liver disease, and the degree of fibrosis is the strongest predictive factor for liver-related mortality (*Anstee et al., 2019*). Current therapies for liver fibrosis focus on eliminating the underlying etiology. However, there is a lack of effective treatment for several chronic liver diseases, such as non-alcoholic fatty liver disease, which affects one in four people worldwide (*Younossi et al., 2016*), and primary sclerosing cholangitis (*Karlsen et al., 2017*). Therefore, there is an urgent need to develop antifibrotic therapies. Activation of HSCs in the setting of chronic liver injury represents a critical event in fibrosis, as activated HSC myofibroblasts are the primary source of collagen production and excessive extracellular matrix deposition (*Friedman et al., 1985*; *Maher and McGuire, 1990*; *Mederacke et al., 2013*). With evidence that the scarring process in the liver is reversible (*Sun et al., 2020*) and that HSC myofibroblasts demonstrate plasticity and can revert to an inactive state (*Kisseleva et al., 2012*; *Troeger et al., 2012*), there is increasing enthusiasm for development of approaches to promote HSC inactivation as a therapeutic strategy to treat liver fibrosis. Although some therapies under investigation in clinical trials, including PPAR agonists and TGF-β inhibitors, are anticipated to promote HSC inactivation, none have yet been recognized as effective antifibrotic agents (*Guo and Lu, 2020*).

Therefore, with the goal to identify new compounds with antifibrotic potential and novel pharmacological targets for the treatment of liver fibrosis, we performed a small molecule compound screen using primary human HSC myofibroblasts. Combining high-content microscopy imaging and high-throughput qPCR screening approaches as well as filtering methods that take into consideration both the potency and diversity of the candidates' chemical structures, we screened 15,867 experimental wells and identified 19 top candidates that are potentially more potent than the previously discovered HSC-inactivating compound nortriptyline (*Chen et al., 2017*). The top 19 compounds were listed in *Table 1*, and the full datasets of the screening results are provided as Supplementary files, which can serve as a resource for future studies. Further studies to investigate the compounds identified by the screen will deepen our understanding of HSC biology and allow identification of additional genes and pathways that could be targeted to reduce liver fibrosis. There are two potential limitations of this screen. First, although lipid droplet re-accumulation is recognized as a feature of quiescent and inactivated HSCs (*Zisser et al., 2021*), it is possible that partially inactivated HSCs may not accumulate enough lipid droplets to pass our primary screen cutoff, and therefore compounds that act to reduce

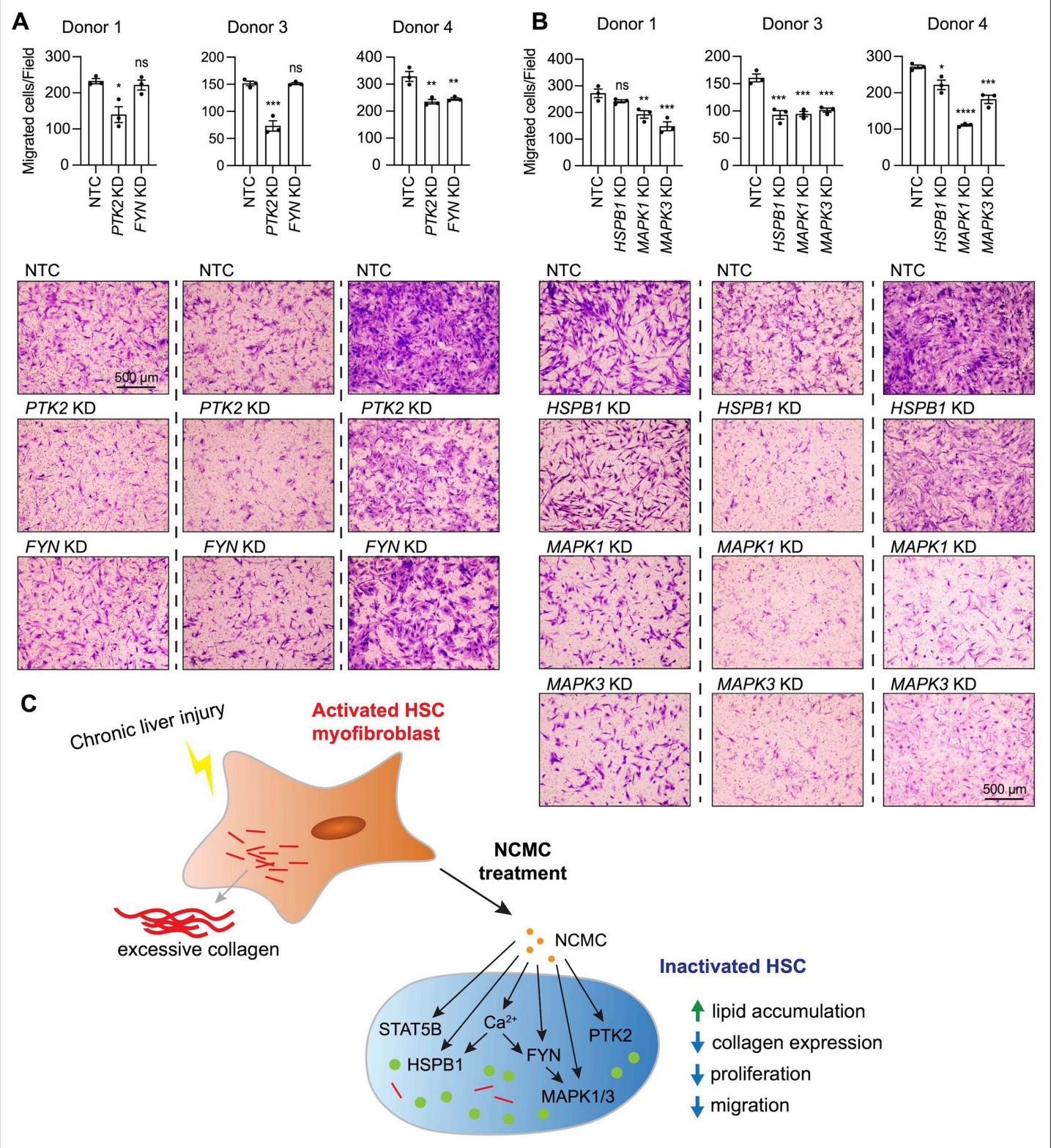

**Figure 7.** MAPK and PTK2 regulate HSC migration.

(**A-B**) Kinases were depleted in HSCs with pooled siRNAs. NTC indicates non-targeting siRNA control. KD indicates the gene transcript that is knocked down. After 72 hr, cells were seeded for transwell migration assay. Migration was assayed at 24 hr for HSCs from donor 1 and 4, and 6 hr for HSCs from donor 3. Top: quantified results. Error bars represent mean ± SEM (n=3). Results are shown for three donor lines. ns indicates not significant (p>0.05), * indicates p<0.05, ** indicates p<0.01, *** indicates p<0.001, and **** indicates p<0.0001 (one-way ANOVA test). Bottom: representative images. Scale

*Figure 7 continued on next page*

*Figure 7 continued*

bar represents 500 μm. At least one experiment was performed independently for each of three donor lines. (**C**) Schematic summarizing the signaling pathways triggered by NCMC treatment and the effect on HSC activity. Red lines represent collagen, orange spots represent NCMC, and green spots represent lipid droplets. This figure has one supplement.

collagen expression or other fibrosis-related aspects of HSCs without affecting lipid accumulation may be missed in this screen. Second, the lipid accumulation and mRNA screens were conducted with primary human HSCs isolated from the same donor 1, therefore we performed the DRC screen in primary human HSCs isolated from donor 2, so that the final screen hits are not restricted to a single HSC isolate. We validated NCMC in primary HSC lines from two additional donors, but for the other positive hits in the screen, further validation in HSC from additional donors is needed.

We focused on NCMC because it strongly induced HSC inactivation, and the activity of NCMC was poorly understood. NCMC belongs to a group of naturally occurring polyether ionophores, which consist of over 120 known members (*Huang et al., 2018*). Among them, the compound monensin shares a similar chemical structure to NCMC and is also a positive hit in our primary screen. It was grouped in the same chemical cluster as NCMC, but NCMC was selected as the representative compound for this cluster in the screen because NCMC had a higher scaled value (*Supplementary file 4*). This group of compounds demonstrates antibacterial, antifungal, antiparasitic, antimalarial, antiviral, anti-inflammatory activities and cytotoxicity in cancer cells (*Kevin Ii et al., 2009*). Although some polyether ionophores have been employed as veterinary antibiotics, none have been used as antibiotics in human, possibly due to concerns about toxicity (*Huczyński, 2012*). Indeed, we observe that NCMC demonstrated substantial cytotoxicity in cell culture at concentrations higher than 10 μM (*Figure 2—figure supplement 1*). However, given that the EC50 of NCMC in lipid accumulation assay is in the range of 10–300 nM without inducing apoptosis, NCMC or compounds with similar structure may have potential as antifibrotic therapy within optimized therapeutic doses.

Despite the extensive study of some polyether ionophores, limited data are available describing the activity and mechanism of action of NCMC in mammalian cells. A screen for bioactive inhibitors of the Otub1/c-Maf axis in multiple myeloma cells demonstrated that NCMC induces c-Maf poly-ubiquitination and proteasomal degradation in the presence of Otub1 (*Xu et al., 2020*). A-130-A, a close analog of NCMC, also inhibited the Wnt/β-catenin pathway and induced autophagy, and both A-130-A and NCMC increased cytosolic $Ca^{2+}$ and reactive oxygen species (ROS) as well as enhanced

**Table 2.** Pharmacokinetics of NCMC in vivo.

Plasma and tissue concentrations of NCMC were measured at the indicated time points after once daily intraperitoneal injection of 1.0 mg/kg NCMC to male C57/B6N mice (n=4) for 4 days.

| Day | Time [h] | Tissue | N | mean [nmol/L] | SD [nmol/L] | CV [%] |
|---|---|---|---|---|---|---|
| 1 | 4 | Plasma | 4 | 109.7 | 23.19 | 21.1 |
| 1 | 24 | Plasma | 4 | 38.25 | 3.18 | 8.3 |
| 2 | 24 | Plasma | 4 | 32.15 | 3.99 | 12.4 |
| 4 | 2 | Plasma | 4 | 69.43 | 2.92 | 4.2 |
| 4 | 4 | Plasma | 4 | 66.70 | 13.77 | 20.6 |
| 4 | 8 | Plasma | 4 | 80.13 | 13.55 | 16.9 |
| 4 | 12 | Plasma | 4 | 67.58 | 12.54 | 18.6 |
| 4 | 24 | Plasma | 4 | 35.25 | 8.97 | 25.4 |
| 4 | 24 | Muscle | 4 | NOP | na | na |
| 4 | 24 | Liver | 4 | NOP | na | na |
| 4 | 24 | Fat | 4 | 375.3 | 145.2 | 38.7 |

N: number of mice, SD: standard deviation, CV: coefficient of variance, NOP: no peak, na: not applicable

the permeability of the mitochondrial inner membrane to $H^+$ and $K^+$ (*Huang et al., 2018*). To examine a wide variety of signaling pathways that may be affected downstream of NCMC, we performed a phospho-kinase array analysis to measure the phosphorylation of 37 kinases at a total of 43 different sites and the total protein expression of β-catenin and HSP60. Seven kinases, including FYN, HSPB1, MAPK1, MAPK3, STAT5A, STAT5B, and PTK2, were selected and further tested because their phosphorylated protein was reduced by NCMC treatment, and all were relatively abundant in HSCs. When depleted individually in HSCs from multiple donors, most of these kinases reduced *COL1A1* level consistently (*Figure 5B*), suggesting that each may play a role in mediating NCMC's effect on collagen expression. It is unclear why STAT5A siRNAs, which effectively depleted *STAT5A* but also reduced *STAT5B* mRNA level, did not significantly affect *COL1A1* expression, considering that reduction of *STAT5B* alone to a similar level by STAT5B siRNAs did demonstrate an inhibitory effect (*Figure 5B–C* and *Figure 5—figure supplement 1*). It is possible that the expression of STAT5B protein is inhibited more efficiently in HSCs transfected with STAT5B siRNAs than those transfected with STAT5A siRNAs despite the similar mRNA levels. STAT5A/B homo- and heterodimers could have different individual DNA-binding specificities (*Maurer et al., 2019*), and the ratio of homo- to heterodimers could also affect transcription of the *COL1A1* gene.

Our investigations of FYN revealed that depletion or inhibition of FYN activity suppresses collagen expression in primary human HSCs and deposition of collagen in the ECM, but regulation via FYN did not explain all the effects observed with NCMC. The regulation of collagen expression by FYN is in agreement with a recent study demonstrating that FYN depletion and inhibition in the presence of TGF-β reduces collagen I expression in immortalized human and rat HSC lines (*Du et al., 2020*). This study also observed that FYN depletion and inhibition in the presence of TGF-β reduced HSC migration (*Du et al., 2020*), however, our data did not show a consistent effect on migration with depletion of FYN. As cell migration is controlled by a complex signaling network, the difference in the basal activity of the signaling pathways up- or down-stream of FYN may account for the observed differences. In contrast, PTK2 and MAPK1/3 depletion showed a more robust and consistent inhibitory effect among all HSC lines tested, suggesting that these kinases may serve as the critical nodes regulating HSC migration. Depletion or inhibition of FYN or MAPK suppresses HSC proliferation (*Du et al., 2020*; *Pagès et al., 1993*), and PTK2 regulates proliferation in many cell types (*Zhou et al., 2019*). Thus, it is likely that modulation of multiple kinases contributes to the anti-fibrotic effect of NCMC, although the involvement of each kinase may vary depending on the cellular context.

We found that NCMC increases cytosolic $Ca^{2+}$. Although it has not been demonstrated experimentally, it is suspected that NCMC may increase cytosolic $Ca^{2+}$ levels by disrupting $Na^+/Ca^{2+}$ exchange (*Huang et al., 2018*). Of note, the $Na^+/K^+$-ATPase inhibitors identified in our screen can also increase cytosolic $Ca^{2+}$ levels (*Tian and Xie, 2008*), and increased cytosolic $Ca^{2+}$ has been observed to inhibit MAPK1/3 in fibroblasts (*Bosch et al., 1998*; *Chuderland et al., 2020*; *Cook et al., 1997*). As a ubiquitous second messenger with wide-ranging physiological roles, cytosolic $Ca^{2+}$ levels may be a key factor mediating the downstream anti-fibrotic activity of NCMC, but further investigations are needed to unravel the complete signaling cascade.

NCMC has been shown to inhibit tumor growth in a model where breast cancer cells were injected subcutaneously and a multiple myeloma subcutaneous xenograft model (*Huang et al., 2018*; *Xu et al., 2020*). Using the same administration method as described by Huang et al. (1 mg/kg NCMC delivered via intra-peritoneal injection) and increasing the dosing frequency to once daily, we were unable to detect NCMC in liver or muscle, while we observed increased concentrations of NCMC in fat compared to plasma. Our in vitro experiments with mouse liver microsomes support metabolization of NCMC in hepatocytes, which may counteract a potential local depot formation in the liver. As NCMC is not detectable in the liver and has a low plasma concentration with high plasma protein binding, we expect HSC exposure to NCMC to be very low. Consequently, chemical modifications of NCMC are needed to come to a drug suitable for the investigation of liver fibrosis in vivo. It is worth screening chemical modifications of NCMC for a more drug-like derivative in the future. Alternatively, molecules similar in structure to NCMC may be evaluated to identify additional compounds with similar activity in HSCs and greater metabolic stability.

Interestingly, salinomycin, another polyether ionophore, has been reported to protect hepatocytes against $CCl_4$-induced oxidative stress and liver injury in vivo (*Kim et al., 2018*). Salinomycin was tested in our primary screen and showed a modest effect on lipid accumulation but did not reach the cutoff

for selection. A pharmacokinetic study of salinomycin revealed rapid metabolism by liver microsomes and showed that CYP3A4 is the major metabolizing enzyme (*Resham et al., 2015*). Concomitant administration of ketoconazole, a selective CYP3A4 inhibitor with salinomycin in rats increased the systemic exposure of salinomycin to seven fold and the $C_{max}$ to three fold (*Resham et al., 2015*). These findings suggest that in vivo pharmacokinetics of polyether ionophores could also be improved via concomitant inhibition of the metabolizing enzyme.

In summary, this study has identified NCMC as a compound that increases cytosolic $Ca^{2+}$ and regulates multiple kinases, including FYN, PTK2, and MAPK1/3 to drive the inactivation of HSC myofibroblasts (*Figure 7C*). Targeting an individual component of this complex network may suppress certain cellular activities and contribute to HSC inactivation, but it may be necessary to synergistically manipulate multiple targets to achieve antifibrotic effects among genetic diversity, as observed with responses in multiple primary human HSC lines. By regulating multiple signaling pathways, NCMC confers a more robust impact than observed with inhibition of only one pathway, and with structural modifications to improve delivery to the liver, similar compounds could represent a more effective strategy to reduce the progression of liver fibrosis.

# Materials and methods

## Key resources table

| Reagent type (species) or resource | Designation | Source or reference | Identifiers | Additional information |
|---|---|---|---|---|
| Gene (*Homo sapiens*) | FYN (dominant negative mutant with Y213A mutation) | Amplified from the Addgene plasmid pRK5 DN-Fyn | Addgene plasmid pRK5 DN-Fyn (#16033) | |
| Cell line (*Homo sapiens*) | HEK-293 | ATCC | CRL-1573 | |
| Biological sample (*Homo sapiens*) | Primary human hepatic stellate cells (donor 1) | Isolated from human nonparenchymal liver cells (NPCs) purchased from Lonza (cat# HUCNP) | Lonza ID: 4105 | Age: 45; Gender: Male; Race: Caucasian |
| Biological sample (*Homo sapiens*) | Primary human hepatic stellate cells (donor 2) | Isolated from human nonparenchymal liver cells (NPCs) purchased from Lonza (cat# HUCNP) | Lonza ID: 4270 | Age: 35; Gender: Male; Race: Caucasian |
| Biological sample (*Homo sapiens*) | Primary human hepatic stellate cells (donor 3) | Lonza (cat# HUCLS) | Lonza ID: 180761 | Age: 57; Gender: Female; Race: Caucasian |
| Biological sample (*Homo sapiens*) | Primary human hepatic stellate cells (donor 4) | Lonza (cat# HUCLS) | Lonza ID: 182821 | Age: 24; Gender: Female; Race: African American |
| Biological sample (*Homo sapiens*) | Primary human hepatic stellate cells (donor 5) | Isolated from human nonparenchymal liver cells (NPCs) purchased from Lonza (cat# HUCNP) | Lonza ID: 4258 | Age: 51; Gender: Male; Race: African American |
| Antibody | Anti-collagen type I (Mouse monoclonal) | Sigma | C2456 | (1:1000) |
| Antibody | Anti-FYN (Rabbit polyclonal) | Cell Signaling Technology | #4023 | (1:1000) |
| Antibody | Anti-phospho-Src family (Rabbit monoclonal) | Cell Signaling Technology | #6943 | (1:1000) |
| Antibody | Anti-phospho-p44/42 MAPK (Rabbit monoclonal) | Cell Signaling Technology | #4370 | (1:1000) |
| Antibody | HRP-β-Actin antibody (Mouse monoclonal) | Santa Cruz Biotechnology | sc-47778 HRP | (1:5000) |
| Antibody | goat anti-rabbit IgG secondary antibody (Goat polyclonal) | Invitrogen | #32460 | (1:3000) |

*Continued on next page*

*Continued*

| Reagent type (species) or resource | Designation | Source or reference | Identifiers | Additional information |
|---|---|---|---|---|
| Antibody | donkey anti-mouse Alexa Fluor 488 secondary antibody (Donkey polyclonal) | Invitrogen | A-21202 | (1:500) |
| Recombinant DNA reagent | pLJM1-eGFP (plasmid) | Addgene | plasmid# 19319 | |
| Recombinant DNA reagent | pRK5 DN-Fyn | Addgene | plasmid# 16033 | |
| Recombinant DNA reagent | pLJM1-DN-FYN (plasmid) | This paper | | This is a plasmid generated by this study for the expression of dominant negative (Y213A) FYN in HSCs using lentivirus system. Refer to the Materials and Methods section for more information. |
| Sequence-based reagent | FYN-forward primer | This paper | PCR primer | This is the forward PCR primer for cloning human *FYN* gene. The sequence is 5′-CAT GCTAGC GCCACC ATGGGCTGTGTGCAAT GTAAGG-3′. Refer to the Materials and Methods section for more information. |
| Sequence-based reagent | FYN-reverse primer | This paper | PCR primer | This is the reverse PCR primer for cloning human *FYN* gene. The sequence is 5′-AGC GAATTC TTACAGGTTTTCACCAGGTT GGTAC-3′. Refer to the Materials and Methods section for more information. |
| Sequence-based reagent | siRNA: non-targeting control | Horizon Discovery | D-001210–05 | siGENOME |
| Sequence-based reagent | Pooled siRNAs: FYN | Horizon Discovery | MQ-003140–04 | siGENOME SMARTpool |
| Sequence-based reagent | Pooled siRNAs: HSPB1/ HSP27 | Horizon Discovery | M-005269–01 | siGENOME SMARTpool |
| Sequence-based reagent | Pooled siRNAs: MAPK3/ ERK1 | Horizon Discovery | M-003592–03 | siGENOME SMARTpool |
| Sequence-based reagent | Pooled siRNAs: MAPK1/ ERK2 | Horizon Discovery | M-003555–04 | siGENOME SMARTpool |
| Sequence-based reagent | Pooled siRNAs: PTK2/FAK | Horizon Discovery | M-003164–02 | siGENOME SMARTpool |
| Sequence-based reagent | Pooled siRNAs: STAT5A | Horizon Discovery | M-005169–02 | siGENOME SMARTpool |
| Sequence-based reagent | Pooled siRNAs: STAT5B | Horizon Discovery | M-010539–02 | siGENOME SMARTpool |
| Sequence-based reagent | siRNA#1: FYN | Horizon Discovery | D-003140–10 | siGENOME |
| Sequence-based reagent | siRNA#2: FYN | Horizon Discovery | D-003140–24 | siGENOME |
| Commercial assay or kit | TaqMan Real-time PCR Assay: Human *ACTA2* | ThermoFisher Scientific | Hs00426835_g1 | |
| Commercial assay or kit | TaqMan Real-time PCR Assay: Human *COL1A1* | ThermoFisher Scientific | Hs00164004_m1 | |
| Commercial assay or kit | TaqMan Real-time PCR Assay: Human *FYN* | ThermoFisher Scientific | Hs00176628_m1 | |
| Commercial assay or kit | TaqMan Real-time PCR Assay: Human *HSPB1/ HSP27* | ThermoFisher Scientific | Hs00356629_g1 | |
| Commercial assay or kit | TaqMan Real-time PCR Assay: Human *MAPK1/ ERK2* | ThermoFisher Scientific | Hs01046830_m1 | |

*Continued*

| Reagent type (species) or resource | Designation | Source or reference | Identifiers | Additional information |
|---|---|---|---|---|
| Commercial assay or kit | TaqMan Real-time PCR Assay: Human *MAPK3/ERK1* | ThermoFisher Scientific | Hs00385075_m1 | |
| Commercial assay or kit | TaqMan Real-time PCR Assay: Human *PSMB2* | ThermoFisher Scientific | Hs01002946_m1 | |
| Commercial assay or kit | TaqMan Real-time PCR Assay: Human *PTK2/FAK* | ThermoFisher Scientific | Hs01056457_m1 | |
| Commercial assay or kit | TaqMan Real-time PCR Assay: Human *STAT5A* | ThermoFisher Scientific | Hs00559643_m1 | |
| Commercial assay or kit | TaqMan Real-time PCR Assay: Human *STAT5B* | ThermoFisher Scientific | Hs00560026_m1 | |
| Commercial assay or kit | TaqMan Real-time PCR Assay: Mouse *Acta2* | ThermoFisher Scientific | Mm00725412_s1 | |
| Commercial assay or kit | TaqMan Real-time PCR Assay: Mouse *Col1a1* | ThermoFisher Scientific | Mm00801666_g1 | |
| Commercial assay or kit | TaqMan Real-time PCR Assay: Mouse *Psmb2* | ThermoFisher Scientific | Mm00449477_m1 | |
| Commercial assay or kit | Proteome Profiler Human Phospho-Kinase Array Kit | R&D Systems | ARY003B | |
| Chemical compound, drug | Nanchangmycin (NCMC) | Selleck Chemicals | S1450 | Used for the initial confirmation of dose response curves in multiple HSC lines |
| Chemical compound, drug | Nanchangmycin (NCMC) | Adooq | A10621 | Used for all other follow-up experiments |
| Chemical compound, drug | 1-Naphthyl PP1 | Tocris | #3603 | |
| Chemical compound, drug | Thapsigargin | Sigma-Aldrich | T9033 | |
| Chemical compound, drug | Ionomycin | Biogems | #5608212 | |
| Software, algorithm | BIOVIA Pipeline Pilot | Dassault Systèmes | | |
| Software, algorithm | GSEA | UC San Diego and Broad Institute | http://www.gsea-msigdb.org/gsea/index.jsp | |
| Software, algorithm | FastQC | Babraham Bioinformatics | v 0.11.8 | |
| Software, algorithm | RSEM | https://github.com/deweylab/RSEM; *Dewey Lab, 2020* | v 1.3.1 | |
| Software, algorithm | Enrichr | https://maayanlab.cloud/Enrichr/ | | |

## Cell culture and compounds

HEK-293 cell line was obtained from ATCC (CRL-1573) and cultured in Dulbecco's Modified Eagle Medium (DMEM) with 10% fetal calf serum (FCS) and 1% Penicillin/Streptomycin (P/S). The cells are mycoplasma negative. Human primary hepatic stellate cells from donors 1, 2, and 5 were isolated from human nonparenchymal liver cells (NPCs) purchased from Lonza (cat# HUCNP) as described previously (*Chen et al., 2017*). Human primary hepatic stellate cells from donors 3 and 4 were purchased as isolated hepatic stellate cells from Lonza (cat# HUCLS). Donor information is listed below.

| Donor | Lonza ID | Age | Gender | Race | BMI |
|---|---|---|---|---|---|
| 1 | 4105 | 45 | M | Caucasian | 24.2 |
| 2 | 4270 | 35 | M | Caucasian | 42.1 |

*Continued on next page*

*Continued*

| Donor | Lonza ID | Age | Gender | Race | BMI |
|---|---|---|---|---|---|
| 3 | 180761 | 57 | F | Caucasian | 23.6 |
| 4 | 182821 | 24 | F | African American | 48.8 |
| 5 | 4258 | 51 | M | African American | 24.5 |

All hepatic stellate cells were cultured in DMEM with 10% FCS and 1% P/S. The primary lipid accumulation screen and secondary mRNA screen were conducted with HSCs from donor 1 at passage 8, the dose response curve screen was conducted with HSCs from donor 2 at passage 8 or 9, and all other experiments were conducted with HSCs from donors as indicated at passage 8–10.

Nanchangmycin (NCMC) was purchased from two sources. The initial confirmation of dose response curves in multiple HSC lines (*Figure 2B* and *Figure 2—figure supplement 1*) were performed with NCMC purchased from Selleck Chemicals (cat# S1450). All other experiments were performed with NCMC purchased from Adooq (cat# A10621) for higher purity. 1-Naphthyl PP1 was purchased from Tocris (cat# 3063). Thapsigargin was purchased from Sigma (cat# T9033). Ionomycin was purchased from Biogems (cat# 5608212). Stock solutions were made with DMSO.

## Animal experiments

Housing of the animals as well as the animal experiments were conducted in accordance with the European Animal Welfare Act (Directive 2010/63/EU) and the derived German Animal Welfare act (TierSchG) and Animal welfare directive (TierSchVersV). The experimental procedures were authorized by the Regierungspräsidium Tübingen as the responsible local German authority under reference number 19–004 G.

## Primary high-throughput lipid accumulation screen

For each of the 5-day screening cycle, cells were plated on day 1 at 1000 cells/well in 30 μl/well of regular complete media (DMEM +10% FCS+1% P/S) in 384-well plates using Multidrop Combi (Thermo). On day 3, 100 nL/well of compounds from the libraries were transferred by a stainless-steel pin array and Seiko compound transfer robot to the assay plates in duplicates. On day 5, the cells were fixed with 4% paraformaldehyde (diluted with DPBS from 16% stock, Electron Microscopy Sciences, cat# 15710) and incubated at room temperature for 15 min. The cells were washed one time with DPBS and incubated with Bodipy 493/503 (0.25 μg/mL, Invitrogen, cat# D3922) and Hoechst (5 μg/mL, Invitrogen, cat# H1399) for 45 min at room temperature. The plates were washed three times with DPBS, and then 50 μl/well DPBS was added before sealing the plates with adhesive foil cover. The plates were imaged using the ImageXpress Micro Confocal (Molecular Devices) at the Institute of Chemistry and Cell Biology (ICCB)-Longwood screening facility. Images of each well were analyzed using MetaXpress software to calculate the percentage of positive cells (the total number of Bodipy-positive cells [cutoff was adjusted for each plate] divided by the total cell count).

We developed a scoring system to rank the strength of a compound in inducing HSC reversion to the inactive phenotype. A score was calculated as follows: 1. Averaged percent positive cells from duplicates was used to calculate the distance from the baseline of the plate (percentile 75%), 2. Toxicity was penalized (the distance from the average number of cells in the compound wells to the number of cells in the nortriptyline wells), and 3. Poor reproducibility was penalized (the error of the two points to the average value of the duplicates). The score was then normalized using nortriptyline and DMSO scores for each plate. A new parameter was calculated termed 'Scaled' with the formula: Scaled = –1*(median (nortriptyline)-score)/abs(median (nortriptyline)-median(DMSO)).

## Consolidation of screening library

Chemical structures of the screening library were consolidated using the data science workflow software BIOVIA Pipeline Pilot. Protonation states of the structures were standardized, and counter ions were eliminated. We used canonical Simplified Molecular Input Line Entry System (SMILES) as a unique linear textual representation of the chemical structure. This way, the initial 15,867 structures could be mapped onto 7696 unique canonical SMILES of which 4329 are represented by a single well in the library and 3367 occur in up to 19 wells. Multiple occurrences of individual canonical SMILES could

be traced to multiple vendors and/or multiple molar concentrations of the individual probes. Using this analysis, the 711 experimental wells defined as hits were determined to represent 464 individual compounds.

## Clustering analysis of primary screen candidates and selection for secondary analysis

Hits were clustered into 102 groups of structurally similar compounds based on Tanimoto similarities using the computational analysis software BIOVIA Pipeline Pilot. The distance to the center of the cluster was calculated for each compound in the cluster using BIOVIA Pipeline Pilot, and the most common structure for each cluster was defined based on this value. The strongest hit with the most common structure for each cluster was selected as the representative for the cluster. Promiscuous bioactive compounds that contain pan assay interference structures (PAINS) (*Baell and Nissink, 2018*), or that we identified as frequent hits in screens at ICCB-L were not included for further analysis, as the exhibited bioactivity may be attributed to interference with specific assay readouts and/or nonspecific, intractable mechanism of action (*Matlock et al., 2018*). Frequent hits were defined as having a positive hit rate of more than 20% in screens performed at ICCB-L or more than 10 total positive hits in the database of ICCB-L screens. One additional compound was removed because the molecular formula was the same as another selected compound, and one compound was removed due to similarity in structure to nortriptyline (*Supplementary file 4*).

## Secondary mRNA screen for cherrypicked small molecules

For each of the 5-day screening cycle, cells were plated in 384-well plates as in primary screen on day 1. Compounds were added on day 3 using a digital non-contact dispenser D300e (Hewlett Packard) in quadruplicate. On day 5, cell lysates for qPCR were prepared using the Cells-to-$C_T$ 1-Step Taqman Kit (Invitrogen, cat# A25603) according to manufacturer's instructions. Briefly, cells were incubated with 25 µL/well lysis buffer (plus DNase) for 5 min at room temperature, and the reaction was stopped by adding 2.5 µL stop solution and incubating for 2 min at room temperature. A total of 2 µL cell lysates were used in the multiplexed qRT-PCRs to measure *ACTA2*, *COL1A1*, and *PSMB2* mRNA levels. To reduce technical variations, the TaqMan probe for endogenous control gene *PSMB2* was VIC-labeled and primer-limited, so that the *PSBM2* probe can be multiplexed with FAM-labeled probe for *ACTA2* or *COL1A1* in the same qRT-PCR. Details for probes are included in the 'qPCR analysis' section. The results were analyzed by fitting the data to the following linear models: Ct_*ACTA2* ~Ct_*PSMB2* +plate + chemical or Ct_*COL1A1*~Ct_*PSMB2* +plate + chemical. Relative fold changes were calculated from the estimate of each chemical as compared to DMSO control.

## Dose response curve screen

The dose response curve screen was performed using an adapted lipid accumulation assay with live human primary HSCs. Briefly, cells were plated at a density of 2500 cells/well in 384-well plates. After 24 hr, compounds were added in duplicate at concentrations from 0.001 to 10 µM. Nortriptyline (10 µM) and DMSO (0.1%) served as controls. Cells were incubated with compounds for 24 hr, followed by treatment with Bodipy (1 µg/ml; ThermoFisher, cat# D3922) and NucLight Rapid Red (final dilution 1:4000; Essen BioSciences, cat# 4725) for an additional 12 hr to stain lipid droplets and nuclei. Fluorescent signals were measured using an Incucyte S3 system.

Bodipy stained area and nuclei count were determined selecting two fields per well. The Bodipy-stained area per nuclei count was calculated per field and the mean was determined. The percentage of lipid accumulation (referred to as % CTL) in response to compound treatment was analyzed as follows: 100 x ((total green area/count [test compound] – total green area/count [mean DMSO]) / (total green area/count [mean Nor] – total green area/count [mean DMSO])). The total green area is a measure of lipid droplet accumulation in µm²/well, and count indicates the cell (nuclei) number per well.

The dose-response measurements were reviewed and scored independently by three researchers based on the following criteria: Priority 1: the percentage of lipid accumulation (% CTL) is increased at 1 µM, the shape suggests a sigmoidal distribution, and at least two concentrations show increased Bodipy staining before the compound becomes toxic; Priority 2: compound treatment at 3.03 µM and 10 µM (highest concentrations) shows increased Bodipy staining, and % CTL is at least 70% at 10 µM;

Priority 3: compound treatment at 3.03 µM and 10 µM (highest concentrations) shows increased Bodipy staining, and % CTL is less than 70% at 10 µM; Priority 4: only treatment at the highest concentration (10 µM) shows increased Bodipy staining; Priority 5: the curve is almost flat (no response).

## Estimation of EC50 for NCMC

HSCs were plated in 384-well plates (*Figure 2B*) or 96-well plates (*Figure 2C–D* and *Figure 2—figure supplement 2*), treated with NCMC at indicated concentrations for time points as specified, and analyzed by lipid accumulation assay similar to the primary screen. Once the percentage of Bodipy-positive cells was determined, the data were fitted into a sigmoidal four parameter logistic model in Graphpad Prism software to estimate the EC50 of NCMC under each condition.

## Fluorescent microscopy

HSCs were seeded in black-wall 96-well plates (Corning, cat# 3603) and treated with NCMC at different concentrations as indicated. After 48 hr, plates were fixed with 4% paraformaldehyde and stained with Bodipy and Hoechst as in the primary lipid accumulation screen. After the last wash, 200 µl/well DPBS was added, and plates were imaged using a Nikon A1R confocal microscope.

## qPCR analysis

qPCR analysis related to depletion of kinase candidates was performed using lysates prepared with the Cells-to-$C_T$ 1-Step Taqman Kit similar to the secondary mRNA screen, except that HSCs were seeded in 96-well plates and 50 µl lysis buffer (with DNase) and 5 µl stop solution were used. For the other qPCR analyses, RNA samples were prepared using TRIzol (Invitrogen, cat# 15596026), and the concentrations were quantified using Qubit 3 fluorometer (Invitrogen) and the Qubit RNA BR Assay Kit (Invitrogen, cat# Q10211) according to manufacturer's instructions. Reverse transcription was performed using iScript gDNA Clear cDNA Synthesis Kit (BIO-RAD, cat# 1725035) with 1 µg total RNA input, and quantitative real-time PCR was performed using TaqMan Universal PCR Master Mix (Applied Biosystems, cat# 4305719) and TaqMan Real-time PCR Assays (ThermoFisher Scientific) for specific genes listed below.

| | Assay ID |
|---|---|
| Human *ACTA2* | Hs00426835_g1 |
| Human *COL1A1* | Hs00164004_m1 |
| Human *FYN* | Hs00176628_m1 |
| Human *HSPB1/HSP27* | Hs00356629_g1 |
| Human *MAPK1/ERK2* | Hs01046830_m1 |
| Human *MAPK3/ERK1* | Hs00385075_m1 |
| Human *PSMB2* | Hs01002946_m1 |
| Human *PTK2/FAK* | Hs01056457_m1 |
| Human *STAT5A* | Hs00559643_m1 |
| Human *STAT5B* | Hs00560026_m1 |
| Mouse *Acta2* | Mm00725412_s1 |
| Mouse *Col1a1* | Mm00801666_g1 |
| Mouse *Psmb2* | Mm00449477_m1 |

## Scar-in-a-jar (Siaj) assay

To test the effect of NCMC on collagen deposition in ECM, HSCs were seeded in black-wall 96-well plates (Corning, cat# 3603) and treated with DMSO or 100 nM NCMC for 48 hr in Ficoll medium, that is complete medium supplemented with 50 µg/ml L-ascorbic acid 2-phosphate sesquimagnesium salt hydrate (Sigma, cat# A8960), 37.5 mg/ml Ficoll-PM70 (Sigma, cat# F2878), 25 mg/ml Ficoll-PM400 (Sigma, cat# F4375), and 5 ng/mL TGF-β (R&D Systems, cat# 240-B-002). To test the effect of *FYN*

depletion on collagen deposition, HSCs were reverse transfected in 96-well plates with 50 nM non-targeting control siRNA or siRNAs against *FYN* (see the 'RNAi-mediated depletion of genes' section for specific information on siRNAs and transfection reagents). After 48 hr, cells were incubated with Ficoll medium for an additional 48 hr.

Cells were then fixed with ice-cold methanol for 2 min on ice, washed one time with DPBS and then incubated with primary antibody against collagen type I in DPBS (1:1000, Sigma, cat# C2456) at 4°C overnight. After three washes with PBS-Tween (0.05% v/v), cells were incubated with donkey anti-mouse Alexa Fluor 488 secondary antibody (1:500, Invitrogen, cat# A-21202) and Hoechst (1:4000) in DPBS at room temperature for 1 hr. Plates were washed three times with PBS-Tween, and after the final wash, 200 µL/well of DPBS was added. Plates were imaged using the ImageXpress Micro Confocal microscope (Molecular Devices) with 10 x Plan Apo lens, and collagen fibers were analyzed using a custom module built within the MetaXpress software.

## Liver spheroid experiment

Liver spheroids were prepared as previously described (*Leite et al., 2016*) except spheroids were formed from primary rat hepatocytes (Lonza, cat# RSCP01) and primary human HSCs. Cells were seeded in ultra-low attachment round bottom 96-well plates (Greiner Bio-One, cat# 650970) at a ratio of 1:2. The cells were incubated in HCM hepatocyte culture media (Lonza, cat# CC-3198) for one day with orbital shaking to allow the generation of liver spheroids. The spheroids were then treated with DMSO or NCMC for 72 hr with or without TGF-β (5 ng/mL). Spheroids were collected, and RNA was extracted to quantify expression of human *COL1A1* and *ACTA2* expression through qPCR.

## RNA sequencing

HSCs were treated with DMSO or 1 µM NCMC for 48 hr. RNA was extracted using RNeasy Mini kit (Qiagen, cat# 74104), followed by quality assessment via Agilent 2,200 Tape Station. Two biological samples were prepared for DMSO treatment and three biological samples were prepared for NCMC treatment, and all samples had an RNA integrity number (RIN) greater than 9. RNA library was prepared using TruSeq Stranded mRNA Library Prep Kit (Illumina, cat# 20020594) and sequenced on a HiSeq2000.

For data analysis, reads were quality assessed using the FASTQC (v 0.11.8) and aligned to the human reference genome (GRCh38_release_37) from GENCODE with Star aligner (v2.7.3) using RSEM (v1.3.1) with default parameters. First, the human reference genome was indexed using the GENCODE annotations (gencode.v37) with rsem-prepare-reference from RSEM software. Next, rsem-calculate-expression was used to align the reads and quantify the gene abundance. The output of rsem-calculate-expression gives separately the read count and transcripts per million (TPM) value for each gene.

## Differential expression analysis

Differential expression analysis was performed using gene read counts with DESeq2 package (v 1.32.0) to produce LFC values and corresponding p-values (FDR) applying a Benjamini–Hochberg correction for multiple testing. The heatmap was created using normalized gene count values from Deseq2, using R gplots package heatmap.2 function with row scaling.

## Gene set enrichment analysis (GSEA)

Gene set enrichment analysis was performed using the GSEA software downloaded from http://www.gsea-msigdb.org/gsea/index.jsp (*Mootha et al., 2003*; *Subramanian et al., 2005*). An expression dataset containing gene name and log2 (fold change) was generated based on the RNA sequencing results and loaded to the software as the input file. The c5.all.v7.4 gene matrix was used as the database of gene sets, and gene sets smaller than 10 or larger than 1000 in size were excluded for the analysis. The canonical HSC gene signature and specific HSC gene signature were obtained from previous publication (*Zhang et al., 2016*), and the liver cirrhosis signature was downloaded from Disgenet database (*Piñero et al., 2020*). Among the 44 genes in the canonical HSC signature, 35 were found in our differential expression list. Among the 122 genes in the specific HSC signature, 97 were found in our differential expression list. Among the 103 genes in the liver cirrhosis signature, 69 were found in our differential expression list. These genes were listed in *Supplementary file 8*.

## Transwell migration assay

HSCs were treated with DMSO or 1 μM NCMC for 48 hr or transfected with siRNAs for 72 hr in six-well plates. HSCs were then trypsinized and counted to seed at 5000–10,000 cells per insert depending on donor and assay duration (Corning, cat# CLS3422) in serum-free DMEM. Complete medium (with 10% FBS) was added to the bottom well to induce cell migration through the pores (diameter: 8 μm) of the membrane at the bottom of the insert. After the indicated assay time, cells were fixed with 4% paraformaldehyde at room temperature for 15 min and stained with crystal violet (1% w/v in 20% methanol, Sigma, cat# C0775) for 1 hr. Inserts were washed with DPBS, and the cells that had not migrated through the pores and remained on the upper side of the membrane were removed with cotton swabs. Images were taken using EVOS XL Core microscope with 10 x lens under brightfield.

## Wound healing assay

HSCs were plated in CytoSelect 24-well wound healing assay plates with inserts (Cell Biolabs, cat# CBA-120) at 400,000 cells/well in complete medium. Eighteen hours after plating, inserts were removed to generate a 0.9 mm wound field, and cells were incubated with complete medium containing DMSO or NCMC for an additional 30 hr. Images were taken using EVOS FL microscope.

## Proliferation assay

HSCs were seeded in black-wall 96-well plates (Corning, cat# 3603) at 3000 cells/well, and 18 hr later, DMSO and NCMC at different concentrations as indicated were added with six replicates. One plate was fixed on each day with 4% paraformaldehyde for five days consecutively and stored at 4 °C until all plates were ready for staining with Hoechst. ImageXpress Micro Confocal microscope (Molecular Devices) was used for taking four images/well with 10 x Plan Apo lens, and MetaXpress software was used for counting the number of nuclei.

## Apoptosis analysis by flow cytometry

HSCs were treated with DMSO or NCMC at indicated concentrations for 48 hr, followed by trypsinization and staining with Annexin V and propidium iodide using Dead Cell Apoptosis Kit (Invitrogen, cat# V13241) according to manufacturer's instructions. Cells were analyzed using FACSAria II (BD Biosciences).

## Quiescence/cell cycle analysis by flow cytometry

HSCs were treated with DMSO or NCMC at indicated concentrations for 48 hr, trypsinized, harvested, washed with DPBS, and resuspended in 0.5 mL DPBS. Cells were fixed by adding 4.5 mL ice-cold 70% ethanol in a drop wise manner while vortexing and were then kept at –20 °C for at least 2 hr. Cells were washed twice with FACS buffer (DPBS supplemented with 2% heated-inactivated filtered fetal bovine serum and 1 mM EDTA) before resuspending in FACS buffer at $1 \times 10^6$ cells/100 μL. Cells were then incubated with Ki-67 antibody (0.25 μg/100 μL, clone SolA15, Invitrogen, cat# 11-5698-82) in the dark for 30 min at room temperature. After incubation, cells were washed twice with FACS buffer, followed by incubation with propidium iodide staining solution (DPBS supplemented with 50 μg/ml propidium iodide (Invitrogen, cat# P3566), 10 μg/ml RNase (Thermo Scientific, cat# EN0531) and 2 mM $MgCl_2$) for another 20 min at room temperature before analysis by FACSAria II (BD Biosciences).

## Calcium measurements

Fluo-4 NW calcium assay starter kit (Invitrogen, F36206) was used to measure cytosolic calcium according to the manufacturer's protocol in the presence and absence of 1 mM EGTA. HSCs were plated on a Corning 96 well UV transparent plate 24 hr prior to analysis. Media was removed and cells were washed with 1 X calcium and magnesium chloride free PBS before adding the dye mix with probenecid with and without EGTA to each well. Plates were covered in aluminum foil and incubated at 37 °C for 30 min. Plates were equilibrated to room temperature for 30 min prior to analysis. Measurements were performed on a Tecan Infinite M Plex M-200 using I-control 2.0 software from Austria GmbH to measure fluorescence intensity with excitation at 494 nm and emission at 516 nm. Readings were performed by loading the plate immediately following treatment with DMSO or compounds at the indicated concentration. All measurements were normalized to time 0 by subtracting the initial value for each well. Based on this approach, the increase in $Ca^{2+}$ levels measured for ionomycin may

be reduced due to a more rapid response to the compound. Measurements were plotted as change in relative fluorescence unit (RFU) at 50% gain on the Y-axis relative to time 0 in seconds on the X-axis using Graphpad Prism 8.4.3.

## Kinase array analysis

HSCs from donor 3 were treated with DMSO or 1 µM NCMC for 1 hr or 18 hr. Cell lysates were prepared and analyzed using the Proteome Profiler Human Phospho-Kinase Array Kit (R&D Systems, cat# ARY003B) according to manufacturer's instructions. Scanned films were quantified using ImageJ.

## RNAi-mediated depletion of gene expression

HSCs were reverse transfected with siRNAs as indicated using Dharmafect-1 transfection reagent (Horizon Discovery, cat# T-2001) according to manufacturer's instructions. For 12-well plates, 60 µL of 1 µM siRNAs were added to 180 µL Opti-MEM (Gibco, cat# 31985070) for the final concentration of 50 nM and then mixed with diluted Dharmafect-1 in Opti-MEM (1.2 µL Dharmafect-1 in 238.8 µL Opti-MEM). After 30 min, HSCs resuspended in transfection medium (DMEM supplemented with 16% FCS) were seeded in the wells containing the siRNA/Dharmafect-1 mixture at 70,700 cells/mL in 720 µL/well medium. Transfection in other plate formats were scaled up or down accordingly based on surface area. Cells were incubated with siRNAs and transfection reagents for 72 hr before analysis, unless indicated otherwise.

The siRNAs used in this study were purchased from Horizon Discovery, including siGENOME non-targeting control siRNA #5 (D-001210–05), siGENOME SMARTpool siRNAs for *FYN* (MQ-003140–04), *HSPB1* (M-005269–01), *MAPK3* (M-003592–03), *MAPK1* (M-003555–04), *PTK2* (M-003164–02), *STAT5A* (M-005169–02) and *STAT5B* (M-010539–02) and individual siGENOME siRNAs for *FYN* (siRNA#1: D-003140–10, siRNA#2: D-003140–24).

## Cloning, lentivirus packaging, and infection

The cDNA encoding the dominant negative Y213A FYN mutant was amplified from the plasmid pRK5 DN-Fyn (gift from Filippo Giancotti, Addgene plasmid # 16033) using the following PCR primers: forward primer: 5'-CAT GCTAGC GCCACC ATGGGCTGTGTGCAATGTAAGG-3'; reverse primer: 5'-AGC GAATTC TTACAGGTTTTCACCAGGTTGGTAC-3'. The amplified PCR product was digested with NheI and EcoRI enzymes and inserted into linearized pLJM1 plasmid (gift from David Sabatini, Addgene, plasmid# 19319). Whole plasmid sequencing was performed to confirm that the DN-FYN sequence was correct.

HEK-293 cells were transfected with pLJM1-eGFP or pLJM1-DN-FYN plasmid together with lenti-virus packing and envelope plasmids pMD2.G (gift from Didier Trono, Addgene plasmid# 12259) and psPAX2 (gift from Didier Trono, Addgene plasmid# 12260) using X-tremeGENE 9 DNA transfection reagent (Roche, cat# 6365779001) according to manufacturer's instructions. For a 10 cm dish of 293 cells, 30 µL X-tremeGENE 9 reagent, 750 ng psPAX2, 250 ng pMD2.G and 1 µg pLJM1-eGFP or pLJM1-DN-FYN were mixed in 500 µL Opti-MEM and incubated for 15 min before added to culture medium in a drop wise manner. Twenty-four hr later, culture medium was changed, and cells were incubated with fresh regular medium for another 24 hr. Conditioned medium containing virus was then collected and filtered through 0.45 µm filters. HSCs were seeded to reach 30–40% confluency after 18 hr and then infected with viruses. Polybrene (Sigma-Aldrich, cat# TR-1003-G) was used at a final concentration of 10 µg/mL to enhance infection efficiency.

## Western blot

Cells were pelleted and lysed with RIPA buffer (150 mM sodium chloride, 1.0% Triton X-100, 0.5% sodium deoxycholate, 0.1% SDS (sodium dodecyl sulfate) and 50 mM Tris, pH 8.0) supplemented with protease inhibitors (Thermo Scientific, cat# 87786) and phosphatase inhibitors (Thermo Scientific, cat# 78420). Cell lysates were centrifuged to remove debris. Protein concentrations were measured using Pierce BCA Protein Assay Kit (Thermo Scientific, cat# 23227). Bolt LDS Sample Buffer (Invitrogen, cat# B0007) and Bolt Sample Reducing Agent (Invitrogen, cat# B0009) were added to cell lysates and the sample mixture was boiled for 10 min before loading. Bolt 4% to 12% Bis-Tris gels (Invitrogen, cat# NW04120BOX) were used for electrophoresis, followed by transferring with iBlot 2 Dry Blotting System (Invitrogen, cat# IB21002S). Membranes were blocked with 1% BSA (Thermo Scientific, cat#

37520, for phospho-Src family) or 3% milk (Lab Scientific, cat# M0841, for other proteins) at room temperature for 1 hr and incubated with primary antibody at 4 °C for 2 days (for FYN) or overnight (for other proteins). Membranes were washed three times with Tris Buffered Saline-Tween (TBST) buffer (Boston BioProducts, cat# IBB-181–6), incubated with secondary antibody for another 1 hr, washed three times with TBST buffer, and then incubated with SuperSignal West Pico PLUS chemiluminescent substrates (Thermo Scientific, cat# 34580) for 5 min before exposure to film (Ece Scientific Co, cat# E3018). The following antibodies were used: FYN antibody (1:1000, Cell Signaling Technology, cat# 4023), phospho-Src family antibody (1:1000, Cell Signaling Technology, cat# 6943), HRP-β-Actin antibody (1:5000, Santa Cruz Biotechnology, cat# sc-47778), phospho-p44/42 MAPK (Erk1/2) (Thr202/Tyr204) antibody (1:1000, Cell Signaling Technology, cat# 4370), and goat anti-rabbit IgG secondary antibody (Invitrogen, cat# 32460).

## Pharmacokinetics (PK) studies

One mg/kg Nanchangmycin (867.1 Da), suspended in saline (0.9%), was dosed intraperitoneally (10 mL/kg) once day or once daily for 4 days in four male mice (C57BL/6 N). Plasma samples for the determination of compound concentrations were taken on day one at 4 hr and 24 hr post injection, on day two at 24 hr and on day four at 2, 4, 8, 12, and 24 hr to generate a full PK profile at the end of the study. In addition, tissue samples from epididymal fat, liver and tibial muscle were collected at the end of the experiment. Tissue homogenization with a Precellys device was essentially done as described (*Cui et al., 2019*). Prior to bioanalysis, plasma and tissue homogenate samples were spiked with internal standard solution and diluted with acetonitrile/methanol for protein precipitation. Quantification was done by means of HPLC-MS/MS at mass transition of 865.5–251.1 Dalton.

## Microsome analysis

The metabolic degradation of the test compound is assayed at 37 °C with pooled liver microsomes from (male/female) mice (CD1). The final incubation volume of 100 µl per time point contains TRIS buffer pH 7.6 at room temperature (0.1 M), magnesium chloride (5 mM), microsomal protein (0.5 mg/ml) and NCMC at a final concentration of 1 µM. Following a short pre-incubation period at 37 °C, the reactions were initiated by addition of β-nicotinamide adenine dinucleotide phosphate, reduced form (NADPH, 1 mM) and terminated by transferring an aliquot into solvent after different time points. Additionally, the NADPH-independent degradation was monitored in incubations without NADPH, terminated at the last time point. The quenched incubations were pelleted by centrifugation (10,000 g, 5 min). An aliquot of the supernatant was assayed by LC-MS/MS for the amount of NCMC. The half-life ($t_{1/2}$ in vitro) is determined by the slope of the semilogarithmic plot of the concentration-time profile. The intrinsic clearance (CL_INTRINSIC) is calculated by considering the amount of protein in the incubation:

CL_INTRINSIC [µl/min/mg protein] = (Ln 2 / (half-life [min] * protein content [mg/ml])) * 1,000.

## Acknowledgements

We thank the staff at the ICCB-Longwood Screening Facility, and especially Jennifer Smith and Richard Siu for their assistance in establishing and performing the small molecule screen. We thank members of the Mullen lab in addition to Raymond Chung for helpful discussions, along with colleagues at Boehringer Ingelheim, including Birgit Goetz, Katja Thode, Elfriede Mueller, Manuela Schuler, Nicola Zimmerman for experimental support, and Carine Boustany and Daniela Moutinho Dos Santos for helpful discussions. We also thank the MGH Next Gen Sequencing Core, MGH Center for Regenerative Medicine Flow Cytometry Facility, and MGH Program in Membrane Biology Microscopy Core for their support. This work was funded through a grant from Boehringer Ingelheim Pharma GmbH Co KG.

## Additional information

### Competing interests

Michael Bieler: MB is an employee of Boehringer Ingelheim. Achim Sauer: AS is an employee of Boehringer Ingelheim. Julia F Doerner: JFD is an employee of Boehringer Ingelheim. Jörg F Rippmann: JFR is an employee of Boehringer Ingelheim. Alan C Mullen: ACM receives funding from Bristol-Myers Squibb and GlaxoSmithKline for unrelated projects and is a consultant for Circ Bio and Third Rock Ventures. The other authors declare that no competing interests exist.

## Funding

| Funder | Grant reference number | Author |
|---|---|---|
| Boehringer Ingelheim | | Alan C Mullen |

Scientists from Boehringer Ingelheim participated in the discussion on study design, data collection and interpretation, and the decision to submit the work for publication. Scientists from Boehringer Ingelheim also performed experiments included in the manuscript.

## Author contributions

Wenyang Li, Alan C Mullen, Conceptualization, Data curation, Formal analysis, Funding acquisition, Investigation, Methodology, Project administration, Resources, Software, Supervision, Validation, Visualization, Writing – original draft, Writing – review and editing; Jennifer Y Chen, Conceptualization, Data curation, Investigation, Methodology, Project administration, Resources, Software, Visualization, Writing – review and editing; Cheng Sun, Achim Sauer, Data curation, Formal analysis, Investigation, Methodology, Resources, Writing – review and editing; Robert P Sparks, Data curation, Formal analysis, Investigation, Methodology, Validation, Writing – review and editing; Lorena Pantano, Data curation, Formal analysis, Investigation, Methodology, Resources, Software, Visualization, Writing – review and editing; Raza-Ur Rahman, Rory Kirchner, David Wrobel, Michael Bieler, Data curation, Formal analysis, Investigation, Methodology, Resources, Software, Writing – review and editing; Sean P Moran, Data curation, Investigation, Methodology, Software, Writing – review and editing; Joshua V Pondick, Data curation, Investigation, Writing – review and editing; Shannan J Ho Sui, Data curation, Funding acquisition, Investigation, Methodology, Resources, Software, Supervision, Writing – review and editing; Julia F Doerner, Jörg F Rippmann, Conceptualization, Funding acquisition, Investigation, Methodology, Project administration, Resources, Supervision, Writing – review and editing

## Author ORCIDs

Wenyang Li http://orcid.org/0000-0002-1110-1099
Alan C Mullen http://orcid.org/0000-0002-4096-3106

## Ethics

Housing of the animals as well as the animal experiments were conducted in accordance with the European Animal Welfare Act (Directive 2010/63/EU) and the derived German Animal Welfare act (TierSchG) and Animal welfare directive (TierSchVersV). The experimental procedures were authorized by the Regierungspräsidium Tübingen as the responsible local German authority under reference number 19-004-G.

## Decision letter and Author response

Decision letter https://doi.org/10.7554/eLife.74513.sa1
Author response https://doi.org/10.7554/eLife.74513.sa2

# Additional files

## Supplementary files

• Supplementary file 1. Small molecule Compound Libraries. This table lists some basic information of the compound libraries screened in this study, including number of compounds, ICCB-L plate numbers, number of concentrations tested for each compound, and total number of experimental wells.

• Supplementary file 2. Primary Screen Results. This table contains the full results of the primary screen, including compound annotation information (columns A-G), stock concentration in the library (column H), and primary screen results (columns I-R).

• Supplementary file 3. 711 Hits from Primary Screen. This table contains the primary screen results of the 711 hits identified from the primary screen. This table was generated by filtering *Supplementary file 2* based on the last column.

• Supplementary file 4. Clustering of Primary Screen Candidates. This table contains the clustering results of the 711 hits identified from the primary screen. Compound structure used for structure-based clustering (column A), clustering information (columns B-E), compound annotations (columns F-P), primary screen results from this study (columns Q-Y), positive counts and total screen counts of a compound in previous screens conducted at ICCB-L (columns Z-AA), clustering and PAINS analyses results (columns AB-AD).

• Supplementary file 5. Secondary Screen Results. This table lists the results of the secondary mRNA screen. Data were normalized to DMSO control. "FC" indicates "fold change".

• Supplementary file 6. DRC Screen Results. This table lists the results of the drug response curve (DRC) screen. Column A annotates the experimental wells, positive control and the five compounds that were not available for testing at the time of screen. Columns C-N contain the scores each researcher gave for each experiment.

• Supplementary file 7. GSEA of Ontology Gene Sets. This table contains the full results of the gene enrichment analysis (GSEA) of ontology gene sets regardless if a pathway is statistically significant or not. The interpretation of the scores can be found on GSEA website (https://www.gsea-msigdb.org/gsea/index.jsp).

• Supplementary file 8. HSC and Liver Cirrhosis Gene Signatures. This table lists the zScore of the genes analyzed for each signature.

• Supplementary file 9. Significantly differentially expressed genes in NCMC-treated cells. This table contains the 1904 mRNAs up-regulated (increase fold change >2, FDR <0.05) and the 2620 mRNAs down-regulated (decrease fold change >2, FDR <0.05) by NCMC treatment.

• Supplementary file 10. Pathway enrichment analysis of NCMC-regulating genes. This table contains the Enrichr pathway analysis results of the up- and down-regulated genes by NCMC in five databases.

• MDAR checklist

## Data availability

Sequencing data have been deposited in GEO under accession code GSE180980. The compound information and screening results have been provided as supplementary files. Original unprocessed scanned films have been provided as a zip file for all figures containing western blot results.

The following dataset was generated:

| Author(s) | Year | Dataset title | Dataset URL | Database and Identifier |
|---|---|---|---|---|
| Li W, Rahman R, Moran S, Mullen AC | 2021 | Nanchangmycin exerts antifibrotic activity by inhibiting collagen production, migration, and proliferation of hepatic stellate cells | https://secure-web.cisco.com/1XcwOPgsJl6KzdAsomS9kPuJBM95bMphgMNgWtcNmQkcNuSmEBxkptCekg9IybyzIeh_DpTJ7rGjmgquni5RtLsQCqN3aOAI6uQ8BcbkWqtqDu7KHGE7fYvy-BenccWeac83j6XO2x-DNquxAyDHQi3Kc3qioLkSOggE6ZTwcsJ-3Xy5g1uAF1LT6hVGFJQAC5qqckX2ByNfGBwIQLQHNDY7vBzHubpP6laxt1tinRzftFkFM9jboU7diyH9xpNQY/https%3A%2F%2Fwww.ncbi.nlm.nih.gov%2Fgeo%2Fquery%2Facc.cgi%3Facc%3DGSE180980 | NCBI Gene Expression Omnibus, GSE180980 |

The following previously published dataset was used:

| Author(s) | Year | Dataset title | Dataset URL | Database and Identifier |
|---|---|---|---|---|
| Chen JY, Zhou C, Pondick JV, Mullen AC | 2017 | Tricyclic Antidepressants Induce Inactivation of Hepatic Stellate Cell (HSC) Myofibroblasts | https://www.ncbi.nlm.nih.gov/geo/query/acc.cgi?acc=GSE78853 | NCBI Gene Expression Omnibus, GSE78853 |

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
