## [Editor Report]

The manuscript by Li et al., identifies the polyether ionophore nanchangmycin as a novel anti-fibrotic compound through a comprehensive chemical library screen. Given the lack of clinically available treatments for liver fibrosis, the anti-activation properties of nanchangmycin could represent a novel therapeutic avenue for the treatment of this disease. These studies pave way for future studies evaluating clinical efficacy of nanchangmycin and modified nanchangmycin compounds in the future.

---

## [Decision Letter]

**Decision letter after peer review:**

Thank you for submitting your article "Nanchangmycin regulates FYN, FAK and ERK to control the fibrotic activity of hepatic stellate cells" for consideration by *eLife*. Your article has been reviewed by 3 peer reviewers, one of whom is a member of our Board of Reviewing Editors, and the evaluation has been overseen by Jonathan Cooper as the Senior Editor. The following individual involved in review of your submission has agreed to reveal their identity: Pau Sancho-Bru (Reviewer #2).

Essential revisions:

The primary concern shared by all three reviewers is the lack of an in vivo model to test the therapeutic efficacy of Nanchangmycin in modulating hepatic fibrosis. Specifically, an in vivo model of hepatic fibrosis to test clinical efficacy of this compound is absolutely essential for consideration of publication.

*Reviewer #1 (Recommendations for the authors):*

This is a thorough investigation of the anti-fibrotic activity of nanchangmycin. The authors utilize multiple different in vitro and ex vivo model systems to show nanchangmycin inhibits proliferation, migration and activation of HSCs. While the authors do a convincing job demonstrating these effects, the lack of an in vivo model to test the therapeutic potential of this compound limits enthusiasm for publication at this moment. For example, would nanchangmycin be successful at reversing CCL4 induced fibrosis in a rodent model? These data are needed to justify the clinical potential of nanchangmycin. Without in vivo data supporting these in vitro/ex vivo findings, the manuscript is too premature in its current state for publication.

*Reviewer #2 (Recommendations for the authors):*

Would be important to provide more information on the data presented in the Supplementary tables and maybe an interpretation guide, so the readers can look for specific compounds.

The readout used in the fist screening (lipid droplet accumulation) is indicative of only some aspects involved in HSC activation, so the authors may be having a bias on the selection of compounds. Would be good to discuss this strategy and limitations.

Only final validation of compounds was performed with several HSC donors. This is a limitation of the design that would need to be discussed.

Not clear if three independent experiments were performed for each donor HSCs. Please clarify.

In the Results section, for the first screening of drugs, do the authors use any special medium to promote the storage of lipids? It is not clear in the methods section.

Why the authors considered highest priority the compounds with DRC<2 and EC50<5uM?

F3: Pattern of reduction of Acta 2 is not a typical response of HSCs and certainly not "dose-dependent effect of NCMC on Acta2" as stated Line 213. Authors should evaluate other activation markers. Moreover, would be important to also include quiescent markers in the analysis.

Authors should perform a more extensive analysis of RNASeq data from figure 3.

Is NCMC affecting cell metabolism and synthetic pathways that may be behind the induction of antifibrogenic activity?

One would expect that out of 15000 molecules there would be other promising antifibrotic compounds. This should be discussed.

The kinase study is not convincing, and differences observed in the array are very modest. Basically, I am not sure about the design of the study, I believe results would be more clear stimulating the cells to enhance activation/migration and then evaluating the effect of NCMC on kinase phosphorylation and effects.

To finally prove the antifibrogenic effect of NCMC on liver fibrosis and increase the impact of the paper, the authors could incorporate a relevant disease model in the study.

*Reviewer #3 (Recommendations for the authors):*

In vivo validation of Nanchangmycin for liver fibrosis models is strongly recommended.

---

## [Author Response]

Reviewer #1 (Recommendations for the authors):This is a thorough investigation of the anti-fibrotic activity of nanchangmycin. The authors utilize multiple different in vitro and ex vivo model systems to show nanchangmycin inhibits proliferation, migration and activation of HSCs. While the authors do a convincing job demonstrating these effects, the lack of an in vivo model to test the therapeutic potential of this compound limits enthusiasm for publication at this moment. For example, would nanchangmycin be successful at reversing CCL4 induced fibrosis in a rodent model? These data are needed to justify the clinical potential of nanchangmycin. Without in vivo data supporting these in vitro/ex vivo findings, the manuscript is too premature in its current state for publication.

We agree that in vivo data would be helpful. We first performed pharmacokinetic studies to estimate the dose necessary to achieve a NCMC concentration in the liver that approaches the effective concentration that inhibits fibrotic activity in HSCs. However, our in vivo PK analysis show that NCMC does not accumulate to detectable levels in the liver, even after four consecutive days of dosing. In addition, hepatic microsome experiments show that hepatocytes can also metabolize NCMC.

Therefore, it highly unlikely that NCMC will be a viable antifibrotic compound without modifications that increase NCMC concentrations in the liver. We discussed in the manuscript that screening of NCMC derivatives and analogs may yield a more-drug like compound with similar activity that could have greater therapeutic potential. Although modifying NCMC or screening of structurally similar compounds is out of scope for this study, our data 1. provide a resource where a large collection of compounds was tested for potential anti-fibrotic activities in primary human HSCs; 2. highlight NCMC as a potent compound that inactivates HSCs and suggest the potential activity of compounds with similar structures; 3. propose that multiple kinases play a role in the regulation of HSC activity, which may need to be targeted simultaneously to achieve sufficient inhibition of HSC’s fibrotic phenotype; and therefore extend our understanding on the regulation of HSC activities and will inspire future studies in the field of liver and fibrosis research.

Reviewer #2 (Recommendations for the authors):Would be important to provide more information on the data presented in the Supplementary tables and maybe an interpretation guide, so the readers can look for specific compounds.

Thank you for this suggestion. We have added descriptions for each of the Supplementary tables in the “List of supplementary tables” section on pages 67-68.

The readout used in the fist screening (lipid droplet accumulation) is indicative of only some aspects involved in HSC activation, so the authors may be having a bias on the selection of compounds. Would be good to discuss this strategy and limitations.

We appreciate this suggestion. The rationale for selecting lipid droplet accumulation as the primary screen assay was added to the manuscript on page 5. A discussion of the limitation of this assay was added to the manuscript on page 17.

Only final validation of compounds was performed with several HSC donors. This is a limitation of the design that would need to be discussed.

Thank you for this suggestion. We have added a discussion to the manuscript on page 17.

Not clear if three independent experiments were performed for each donor HSCs. Please clarify.

Instead of performing multiple independent experiments in one donor HSC for all experiments, where indicated, we performed one experiment (with replicates) in multiple donor HSCs independently to give us more confidence that our conclusions are generalizable. Given that primary human HSCs can only be cultured for limited passages, it can be difficult to perform three independent experiments for each of multiple HSCs. We have adjusted the descriptions of these experiments in the figure legends to provide more clarity.

In the Results section, for the first screening of drugs, do the authors use any special medium to promote the storage of lipids? It is not clear in the methods section.

Regular culturing medium (DMEM + 10% FCS + 1% P/S) was used for the screen. The method section has been revised to clarify this on page 22.

Why the authors considered highest priority the compounds with DRC<2 and EC50<5uM?

These are selected cutoffs to help us focus on the most potent compound candidates from the screen. We have revised our description on page 7.

F3: Pattern of reduction of Acta 2 is not a typical response of HSCs and certainly not "dose-dependent effect of NCMC on Acta2" as stated Line 213. Authors should evaluate other activation markers. Moreover, would be important to also include quiescent markers in the analysis.

*Acta2* level is usually more sensitive to NCMC response, and its reduction may have already reached the limit around 13.7-41.2nM, therefore we did not observe a dose-dependent effect at higher concentrations. We have revised the Results section on page 9 to describe the observations more accurately. We also added a supplementary figure (Figure 3 —figure supplement 2) demonstrating that the mRNA level of the classic quiescent/inactivation marker PPARγ is induced with NCMC treatment.

Authors should perform a more extensive analysis of RNASeq data from figure 3.Is NCMC affecting cell metabolism and synthetic pathways that may be behind the induction of antifibrogenic activity?

We appreciate this suggestion. We performed a more extensive analysis of RNASeq data as suggested and added new supplementary figures and tables to the manuscript (page 10, Figure 3 —figure supplement 3, Supplementary Tables 9 and 10). Metabolism and synthetic pathways were among the significantly enriched pathways.

One would expect that out of 15000 molecules there would be other promising antifibrotic compounds. This should be discussed.

Thank you for the suggestion. There are more compounds that are potentially effective than the 19 compounds that we listed in table 1. We highlighted these 19 compounds in the manuscript because they demonstrated a better DRC score than our positive control compound nortriptyline. The full screen results are provided as supplementary tables to allow others to filter based on their need to identify other promising compounds. We have also improved labeling in these supplemental tables to make them more user friendly as discussed above. We have revised in the Discussion section on page 17 to address this.

The kinase study is not convincing, and differences observed in the array are very modest. Basically, I am not sure about the design of the study, I believe results would be more clear stimulating the cells to enhance activation/migration and then evaluating the effect of NCMC on kinase phosphorylation and effects.

HSCs are spontaneously activated when cultured on regular plastic cell culture dishes. Therefore, we did not further stimulate the cells to enhance activation/migration. We agree that the differences observed in the array are modest, and this is why we validated the function of these identified candidates with further experiments (e.g., knockdown experiments) in HSCs from multiple donors.

To finally prove the antifibrogenic effect of NCMC on liver fibrosis and increase the impact of the paper, the authors could incorporate a relevant disease model in the study.

We agree that in vivo data would be helpful. We first performed pharmacokinetic studies to estimate the dose necessary to achieve a NCMC concentration in the liver that approaches the effective concentration that inhibits fibrotic activity in HSCs. However, our in vivo PK analysis show that NCMC does not accumulate to detectable levels in the liver, even after four consecutive days of dosing. In addition, hepatic microsome experiments show that hepatocytes can also metabolize NCMC.

Therefore, it highly unlikely that NCMC will be a viable antifibrotic compound without modifications that increase NCMC concentrations in the liver. We discussed in the manuscript that screening of NCMC derivatives and analogs may yield a more-drug like compound with similar activity that could have greater therapeutic potential. Although modifying NCMC or screening of structurally similar compounds is out of scope for this study, our data 1. provide a resource where a large collection of compounds was tested for potential anti-fibrotic activities in primary human HSCs; 2. highlight NCMC as a potent compound that inactivates HSCs and suggest the potential activity of compounds with similar structures; 3. propose that multiple kinases play a role in the regulation of HSC activity, which may need to be targeted simultaneously to achieve sufficient inhibition of HSC’s fibrotic phenotype; and therefore extend our understanding on the regulation of HSC activities and will inspire future studies in the field of liver and fibrosis research.

Reviewer #3 (Recommendations for the authors):

In vivo validation of Nanchangmycin for liver fibrosis models is strongly recommended.

We agree that in vivo data would be helpful. We first performed pharmacokinetic studies to estimate the dose necessary to achieve a NCMC concentration in the liver that approaches the effective concentration that inhibits fibrotic activity in HSCs. However, our in vivo PK analysis show that NCMC does not accumulate to detectable levels in the liver, even after four consecutive days of dosing. In addition, hepatic microsome experiments show that hepatocytes can also metabolize NCMC.

Therefore, it highly unlikely that NCMC will be a viable antifibrotic compound without modifications that increase NCMC concentrations in the liver. We discussed in the manuscript that screening of NCMC derivatives and analogs may yield a more-drug like compound with similar activity that could have greater therapeutic potential. Although modifying NCMC or screening of structurally similar compounds is out of scope for this study, our data 1. provide a resource where a large collection of compounds was tested for potential anti-fibrotic activities in primary human HSCs; 2. highlight NCMC as a potent compound that inactivates HSCs and suggest the potential activity of compounds with similar structures; 3. propose that multiple kinases play a role in the regulation of HSC activity, which may need to be targeted simultaneously to achieve sufficient inhibition of HSC’s fibrotic phenotype; and therefore extend our understanding on the regulation of HSC activities and will inspire future studies in the field of liver and fibrosis research.